# The Second Quantum Revolution: Unexplored Facts and Latest News

**Kimberly Intonti** [1], **Loredana Viscardi** [1,*], **Veruska Lamberti** [1], **Amedeo Matteucci** [2], **Bruno Micciola** [1], **Michele Modestino** [1] and **Canio Noce** [1,3]

[1] Department of Physics 'E.R. Caianiello', University of Salerno, 84084 Fisciano, SA, Italy; kintonti@unisa.it (K.I.); velamberti@unisa.it (V.L.); bmicciola@unisa.it (B.M.); mmodestino@unisa.it (M.M.)
[2] Department of Mathematics, University of Salerno, 84084 Fisciano, SA, Italy; amatteucci@unisa.it
[3] CNR-SPIN, University of Salerno, Fisciano, 84084 Fisciano, SA, Italy; cnoce@unisa.it
[*] Correspondence: lviscardi@unisa.it

**Abstract:** The Second Quantum Revolution refers to a contemporary wave of advancements and breakthroughs in the field of quantum physics that extends beyond the early developments of Quantum Mechanics that occurred in the 20th century. One crucial aspect of this revolution is the deeper exploration and practical application of quantum entanglement. Entanglement serves as a cornerstone in the ongoing revolution, contributing to quantum computing, communication, fundamental physics experiments, and advanced sensing technologies. Here, we present and discuss some of the recent applications of entanglement, exploring its philosophical implications and non-locality beyond Bell's theorem, thereby critically examining the foundations of Quantum Mechanics. Additionally, we propose educational activities that introduce high school students to Quantum Mechanics by emphasizing entanglement as an essential concept to understand in order to become informed participants in the Second Quantum Revolution. Furthermore, we present the state-of-art developments of a largely unexplored and promising realization of real qubits, namely the molecular spin qubits. We review the available and suggested device architectures to host and use molecular spins. Moreover, we summarize the experimental findings on solid-state spin qubit devices based on magnetic molecules. Finally, we discuss how the Second Quantum Revolution might significantly transform law enforcement by offering specific examples and methodologies to address the evolving challenges in public safety and security.

**Keywords:** Second Quantum Revolution; quantum technology; entanglement; modern physics; molecular chemistry; quantum cryptography

## 1. Introduction

The evolution of Quantum Mechanics (QM) is a fascinating story that spans the 20th century and leads to revolutionary discoveries and fundamental technological applications [1]. Here, we present a summary of the key facts and concepts of QM from the beginning up to the current day:

1. Planck's Quantum Theory (1900) [2]: the starting point of QM was Planck's proposal that the energy of harmonic oscillators, such as those emitting electromagnetic radiation, cannot take any value save for discrete values called quanta.
2. Bohr's Atomic Model (1913) [3]: Bohr extended these concepts to atomic structure, proposing a model where electrons have quantized angular momentum and travel on quantized orbits.
3. Schrödinger's Wave Equation (1925) [4]: Schrödinger developed the wave equation, a fundamental equation describing the time-dependent evolution of a quantum state.
4. Heisenberg's Uncertainty Principle (1927) [5]: Heisenberg formulated the Uncertainty Principle, stating that it is impossible to simultaneously know with precision both the position and linear momentum of a particle.

5. Einstein–Podolsky–Rosen Paradox (EPR) (1935) [6]: the EPR Paradox is based on quantum entanglement, a state where the properties of two particles are closely correlated even at large distances, thereby highlighting implications on the nature of reality.
6. Bell inequality (1964) [7]: Bell formulated an inequality to establish limits on the correlations that can exist between measurements of entangled particles.

From its inception to current developments, QM has gone through various stages, culminating in the so-called Second Quantum Revolution [8], which is where quantum entanglement is the key feature, with advanced and promising applications that are shaping the future of technology and science. The Second Quantum Revolution represents an epochal phase in the evolution of technology, with significant impacts across various sectors. At the heart of this revolution lies quantum computing, a mode of information processing that harnesses the principles of QM to outperform the most powerful classical computers in solving specific problems. Specifically, a quantum device is able to perform certain computational tasks in a feasible amount of time while the same task would have been impractical or impossible for even the most advanced classical supercomputers. However, this quantum advantage does not mean that quantum computers will replace classical computers for all tasks. Instead, it highlights their potential to excel in specific computations, such as complex simulations or cryptography-related problems. Despite achieving quantum advantage in certain contexts, the practical implementations of quantum computers for widespread use are still in the early stages of development, and ongoing research has focused on refining the technology and exploring its full capabilities [9].

One of the most revolutionary aspects is the use of quantum bits (qubits), quantum information units that can exist in a state of superposition, which allow the simultaneous handling of multiple possibilities. This characteristic makes quantum computing particularly effective in solving complex problems, such as factoring very large numbers or simulating molecular systems for the development of new drugs. However, the scope of the Second Quantum Revolution extends beyond computation. It reaches into sectors like cryptography, with the threat posed by quantum computers to current cryptographic security techniques, and quantum metrology promising levels of precision previously thought unattainable. Furthermore, the Second Quantum Revolution paves the way for new frontiers in quantum communication, with the possibility of transmitting information completely securely through the indestructibility characteristic of quantum states.

Since a plethora of the scientific literature has been dedicated to the Second Quantum Revolution [10], in this work, we present some unexplored facts and the latest news on this topic. Specifically, we discuss, in the next section, how the entanglement works in the physics realm and its philosophical implications. In Section 3, we outline a way to introduce the entanglement in modern physics teaching, while Section 4 is devoted to the description of a new avenue for quantum technologies, which is represented by the application of quantum molecular materials to quantum computing. Section 5 contains a discussion on harnessing quantum complexity and its implications for potential industrial applications, while Section 6 is devoted to the conclusions.

## 2. Entanglement: From Philosophy to the Basic Concept of Physics

"[. . . ] that one body may act upon another at a distance through a vacuum without the mediation of any thing else by & through which their action or force may be conveyed from one to another is to me so great an absurdity that I believe no man who has in philosophical matters any competent faculty of thinking can ever fall into it." Isaac Newton [11].

At the beginning of the 20th century, Classical Physics was deeply in crisis. This crisis became evident with the study of the black body spectrum, photoelectric effect, and light diffusion from electrons (Compton Scattering): it emerged that the principal contradiction in Classical Mechanics lies in the conceptual separation between particles, with a finite number of degrees of freedom, as well as fields, with an infinite number of degrees of freedom. Indeed, as it is well known, the classical description makes the stability of matter interacting with

radiation very problematic, if not impossible (no model based on classical electrodynamics can reproduce the dynamical equilibrium that exists between the charged particles in an atom) [12].

QM, with its relativistic extensions including quantum field theories, can solve these problems in a very natural way and, up to now, it has represented the best theory we have to describe phenomena at atomic and subatomic scales. Nevertheless, the interpretations of QM are problematic, since "even a century after its initial development, its consequences for our view of reality are still the subject of controversial discussions" [13]. The link between different measurements and corresponding classical intuitions about the impact of a measurements is a recent object of analysis [14]. This study entails giving each mutually orthogonal set of measurement outcomes a measuring context, noting which outcomes are shared by many contexts and deriving relationships between the contexts based on the shared outcomes. It is not our aim here to discuss in detail all of the interpretations of QM or even to list them all; it may be worth noting that some physicists maintain that the so-called Multi World Interpretation (MWI) naturally explains the non-local properties of entanglement [15]; however, it does with a cost (see, for example, Chapter 4 of [16]). Therefore, there is still no interpretation that can be singled out as the best beyond matters of personal preference. In the following, we will use the term 'Copenhagen' to describe the early interpretation by Bohr, which posits that physical systems do not have definite properties until they are measured.

Here, we will review the main phases of that discussion, which lasted several years and continues to this day, and how the main concepts that we will present have emerged from it.

### 2.1. Einstein's Attacks

Let us briefly review the postulates of QM.

Postulate I

The pure states of a physical system $\mathcal{S}$ are in a biunivocal correspondence with the radii of Hilbert space $\mathcal{H}_{\mathcal{S}}$, which is complex and separable.

Postulate II

Each observable $A$ of $\mathcal{S}$ is represented by a linear and self-adjoint operator $\hat{A}$ on $\mathcal{H}_{\mathcal{S}}$.

Postulate III

If $|\psi\rangle$ is a vector of $\mathcal{H}_{\mathcal{S}}$ and represents a pure state $\xi$ of the system $\mathcal{S}$ at the time of the measurement of an observable $A$, then the probability of obtaining a possible outcome $a_n$ is

$$P(a_n) = |\langle u_n|\psi\rangle|^2, \tag{1}$$

where $|u_n\rangle$ is the eigenstate that verifies the eigenvalue equation

$$\hat{A}|u_n\rangle = a_n|u_n\rangle . \tag{2}$$

Postulate IV

The temporal evolution of an isolated system $\mathcal{S}$, from state $|\psi(t_0)\rangle$ to a state $|\psi(t)\rangle$, is realized by a unitary transformation $U(t, t_0)$ as follows:

$$|\psi(t)\rangle = U(t, t_0)|\psi(t_0)\rangle , \tag{3}$$

where the non-relativistic case $U(t, t_0)$ verifies the Schrödinger equation

$$i\hbar\frac{dU}{dt} = HU , \tag{4}$$

with $H$ being the Hamiltonian of the system.

In the Copenhagen interpretation, the vector state $|\psi\rangle$ provides the most complete description of a quantum system. From it, we can obtain, according to Postulate III, the

probability of measuring a given outcome for any observable, as defined in Postulate II. The most complete characterization of a system in the quantum regime is obtained from the calculation of a complete set of observables. The point is that, contrary to what happens in Statistical Mechanics, the probability represents the intrinsic nature of the object described in QM. It is not a tool used as a consequence of our ignorance about variables and parameters that characterized our system.

From the first three postulates, one can derive, as detailed in [12], the famous Uncertainty Relations between the non-commutative operators (or incompatible observables) $\hat{A}$ and $\hat{B}$:

$$\Delta\hat{A}\Delta\hat{B} = \frac{1}{2}|\langle\psi|\,[\hat{A},\hat{B}]\,|\psi\rangle|,\tag{5}$$

where $\Delta\hat{O}$ denotes the standard deviation of the operator $O$, i.e., $\sigma_{\hat{O}} = \sqrt{\langle\hat{O}^2\rangle - \langle\hat{O}\rangle^2}$.

So, in QM, not only is it impossible to measure incompatible physical quantities with infinite precision, but the realities of those two observables are also mutually exclusive. This aspect, apparently contradictory, has its origin in what Bohr defined the Complementarity Principle. The Complementarity Principle asserts that the two aspects of duality, corpuscular and undulatory (through which a quantum mechanical system manifests itself), cannot be observed simultaneously.

Bohr revealed the concept of Complementarity in 1927 at the international conference (held in Como, Italy) in honor of Alessandro Volta. However, Einstein did not participate in that congress. The two physicists had the opportunity to meet and begin the historical clash the following month, in Bruxelles during the 5th Solvay Congress.

Einstein never accepted the claimed completeness of QM and, in a first phase, rejected the Uncertainty Principle. Indeed, he contrived several *Gedankenexperiment* to prove that it is possible to include simultaneous measurements of incompatible variables such as momentum and position. One of the most famous is the following.

Consider three screens: the first with one slit, the second with two slits, and the third acting as the revealing screen, as shown in Figure 1. Einstein hypothesized that, by observing the momentum gained by the first screen in the collision with a particle, it is possible to find the direction of deflection of the latter and to identify which of the two slits had been crossed by the particle. By repeating the experiment with a large number of particles, one would be able to simultaneously see the wave-like and particle-like behavior in contradiction with the Complementarity Principle. Bohr easily demonstrated that this approach is inconsistent, because the error of the measure of the momentum is $\Delta p < ha/\lambda l$ due to the Uncertainty Relations, where $a$ is the distance between the two first screens, $l$ is the distance between the two slits, and $\lambda$ is the particle wavelength. This implies such an error for the screen position $\Delta x > \lambda l/a$ that it cancels the interference pattern. The precision of measurements of the two incompatible observables is mutually exclusive and all of Einstein's attempts failed when formulated in this direction.

Nevertheless, he did not give up. In the early 1930s, Einstein seemed to be unconvinced of the logical consistency of Quantum Theory, but remained convinced about its incompleteness. Then, in May 1935, he published, with Podolsky and Rosen, one of the most important works in the history of physics [6]: "Can Quantum-Mechanical Description of Physical Reality Be Considered Complete?"

The three authors introduced the work with the following abstract:

*"In a complete theory there is an element corresponding to each element of reality. A sufficient condition for the reality of a physical quantity is the possibility of predicting it with certainty, without disturbing the system. In QM in the case of two physical quantities described by non-commuting operators, the knowledge of one precludes the knowledge of the other. Then either (1) the description of reality given by the wave function in QM is not complete or (2) these two quantities cannot have simultaneous reality. Consideration of the problem of making predictions concerning a system on the basis of measurements made on another system that had previously interacted with it*

*leads to the result that if (1) is false then (2) is also false. One is thus led to conclude that the description of reality as given by a wave function is not complete."*

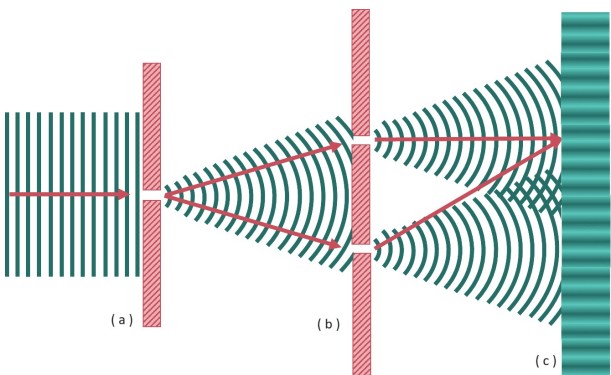

**Figure 1.** Einstein's double-slit *Gedankenexperiment*. The first screen (**a**), on the left, contains only one slit. Here, the particle undergoes a change in momentum due to the collision. The second screen (**b**) contains two slits, as in the original Young experiment. The third screen (**c**), on the right, is the revealing screen.

Reading the article, one realizes that the main idea is based on three principles:

1. Reality:
   If, without in any way disturbing a system, we can predict with certainty (i.e., with probability equal to unity) the value of a physical quantity, then there exists an element of physical reality corresponding to this physical quantity."
2. Completeness:
   "[...] every element of the physical reality must have counterpart in the physical theory."
3. Locality:
   "On the other hand, since at the time of measurement the two systems no longer interact, no real change can take place in the second system in consequence of anything that may be done to the first system."

Einstein, Podolsky, and Rosen considered a two-particle system that, after a certain time interval $T$, separates. Let $\Psi(x_1, x_2)$ be the wave function that completely describes the system. After a period of time $t > T$, we want to measure an observable $A$ on Particle II, so that the state can be written as follows:

$$\Psi(x_1, x_2) = \sum_n \psi_n(x_1) u_n(x_2), \tag{6}$$

where $u_n$ are the eigenfunctions of $\hat{A}$ that we assume to have a discrete spectrum $a_n$. If the result of a measurement is $a_k$, then the system will collapse into the following state:

$$\psi_k(x_1) u_k(x_2). \tag{7}$$

Let $B$ be another observable characterized by the discrete spectrum $b_n$ with eigenfunctions $v_n$. Thus, we can express the state as follows:

$$\Psi(x_1, x_2) = \sum_n \phi_n(x_1) v_n(x_2). \tag{8}$$

If the observable $B$ is measured and $b_r$ is found as result, the system will collapse into the following state:

$$\phi_r(x_1) v_r(x_2). \tag{9}$$

It follows that the state of the first particle collapses into two different states as a consequence of the measurement on Particle II. Anyway, based on Principle (3), there

should be no difference between the two cases from the point of view of Particle I. Moreover, if $A$ and $B$ are two incompatible observables, the two eigenfunctions of Particle I, $\psi_k(x_1)$, and $\phi_r(x_1)$ must represent the same physical reality of the two incompatibles variables, which is in contrast with the mutually exclusive realities. This indicates that there is a false element between the assumptions. EPR identified this element in the hypothesis that considered the wave function a complete description of physical reality and concluded the following:

> "While we have thus shown that the wave function does not provide a complete description of the physical reality, we left open the question of whether or not such a description exists. We believe, however, that such a theory is possible."

### 2.2. To Bohr or Not to Bohr

Bohr's answer was published after a few months, and the author provocatively chose for his work the same title used by Einstein, Podolsky, and Rosen. Bohr did not question the correctness of the argument presented by the three physicists; instead, he claimed that the EPR paradox revealed the essential inadequacy of the classical interpretation on the phenomena explained by QM. In his mind, all measurements can only be prepared, performed, and expressed in a classical way, and physics concerns only the observational acts and the relations between them. In this picture, the mutual interaction between the objects and the measuring apparatus makes the separation between their properties impossible. This is nothing more than the definition of observables in QM: an operator that, as defined in Postulate II, acts on a state as defined in Postulate I. It is impossible to consider the properties of an object as real separately from the act of measuring them. From Bohr's point of view, in the EPR conclusion, there is an essential misunderstanding: simultaneously measuring two complementary variables means to use a mutually exclusive experimental set, i.e., two mutually exclusive acts of observations. It is impossible to assign any element to reality without measurements. Thus, QM is complete if we accept the idea that a system does not possess the properties that cannot be measured. The answer from Einstein, Podolsky, and Rosen to this counterattack was already present in their article:

> "One could object to this conclusion on the grounds that our criterion of reality is not sufficiently restrictive. Indeed, one would not arrive at our conclusion if one insisted that two or more physical quantities can be regarded as simultaneous elements of reality only when they can be simultaneously measured or predicted. On this point of view, since either one or the other, but not both simultaneously, of the quantities P and Q can be predicted, they are not simultaneously real. This makes the reality of P and Q depend upon the process of measurement carried out on the first system, which does, not disturb the second system in any way. No reasonable definition of reality could be expected to permit this."

In conclusion, Einstein, Podolsky, and Rosen rejected Bohr's point of view because its acceptance would have meant admitting the possibility of contradicting the principle of locality, i.e., admitting the existence of something that can communicate instantly, which violates the principle of causality [17].

### 2.3. The Bell Sentence

Not only Einstein, but many other physicists were skeptics with respect to the non-deterministic and non-local aspects of QM. They believed that the intrinsic probabilistic nature of Quantum Theory was due to our incomplete knowledge of properties and the dynamical variables of the individual systems. After all, the same conclusion of Einstein, Podolsky, and Rosen on the incompleteness of QM represented a clue to the existence of hidden classical variables that, to which we do not have access, determined the properties of the systems and the relative outcomes before the measure. Various physicists over the years have elaborated theories that include such local hidden variables, but none of these can reproduce the results of QM. This is not the case of non-local hidden variable theories

like the Bohm model [18]. John Bell, in 1964, demonstrated definitively that no local hidden variable theories can reproduce the statistical predictions of QM [7].

Bell Theorem: No local deterministic hidden variable theory can reproduce all of the predictions of QM. Central to Bell's theorem is the Bell inequality, which imposes constraints on the correlations that can arise between measurements performed on entangled quantum particles. Recent advancements in experimental physics have demonstrated violations of the Bell inequality [19], as highlighted by the groundbreaking experiments recognized by the 2022 Nobel Prize in Physics [20]. In addition to confirming the theoretical predictions of Bell's theorem, the experimental verification of breaking the Bell inequality highlights the non-classical character of QM and its divergence from local realism, which has important implications for our comprehension of the underlying ideas guiding particle behavior at the quantum level.

*2.4. Back to Duality*

It seems we have to accept the idea that the nature of the "very small world", where QM works, is non-deterministic and non-local. It may be asked how these characteristics are related to the duality aspect of that world, which is expressed by the Bohr Complementarity Principle and is the object of the historical debate seen in the previous paragraph. The mathematical quantification and formulation of the duality concept was introduced by Wootters and Zurek in 1979 in [21], where the authors defined the notions of the wave-like and particle-like features of a quantum system. The quantities associated to these concepts, where visibility $V$ corresponded to the wave-like features and the predictability $P$ corresponded to the particle-like features, were shown to verify the following inequality [22]:

$$P^2 + V^2 \leq 1. \tag{10}$$

More recently, it has been discovered that entanglement plays a crucial role in understanding duality; in particular, it turns out that, in a bipartite system, entanglement—which is expressed in terms of concurrence $C$—verifies the triality relation as follows [23]:

$$P^2 + V^2 + C^2 = 1. \tag{11}$$

In order to apply these concepts to the Einstein double-slit *Gedankenexperiment*, we can consider a general two-qubit state. Firstly, we underline that we can mathematically represent a qubit as a linear combination of its basis states, which is typically denoted as $|0\rangle$ and $|1\rangle$. Thus, a qubit state can be written as follows:

$$|\psi\rangle = \alpha |0\rangle + \beta |1\rangle, \tag{12}$$

where $\alpha$ and $\beta$ are complex numbers representing probability amplitudes, and $|0\rangle$ and $|1\rangle$ are the basis states. These complex probability amplitudes satisfy the following normalization condition:

$$|\alpha|^2 + |\beta|^2 = 1. \tag{13}$$

Then, we can consider a general two-qubit state in the qubit basis $|00\rangle, |01\rangle, |10\rangle, |11\rangle$ as follows:

$$|\psi\rangle_{bit} = \alpha_0 |00\rangle + \alpha_1 |01\rangle + \alpha_2 |10\rangle + \alpha_3 |11\rangle. \tag{14}$$

This is useful to investigate the duality in the context of the Young double-slit experiments, where the absence and the presence of the photon in each path can be encoded by $|0\rangle$ and $|1\rangle$, respectively. As reported in [24], the visibility can be determined by the following equation:

$$V = \frac{p_d^{max} - p_d^{min}}{p_d^{max} + p_d^{min}}, \tag{15}$$

where $p_d$ represents the probability of detecting the system in a specific state. In terms of the coefficients in Equation (14), as in [25], the visibility results in

$$V = 2|\alpha_2^* \alpha_0 + \alpha_3^* \alpha_1| . \tag{16}$$

Similarly, the predictability can be expressed as in [26] as follows:

$$P = \frac{|p_0 - p_1|}{|p_0 + p_1|} , \tag{17}$$

where $p_0$ and $p_1$ are the probabilities of finding the first particle in the state $|0\rangle$ and $|1\rangle$, respectively. From this equation, it follows that the particle-like feature can be expressed as follows:

$$P = \left| \left( |\alpha_0|^2 + |\alpha_1|^2 \right) - \left( |\alpha_2|^2 + |\alpha_3|^2 \right) \right| . \tag{18}$$

On the other hand, the entanglement can be quantified in terms of concurrence ($C$), as defined in [27], as follows:

$$C(\rho) = \max(0, \lambda_1 - \lambda_2 - \lambda_3 - \lambda_4) , \tag{19}$$

where $\rho$ is the density matrix of the system and the $\lambda_i$, in decreasing order, are the square roots of the eigenvalues of the following matrix:

$$R = \rho_{s_1 s_2} \tilde{\rho}_{s_1 s_2} , \tag{20}$$

with $\tilde{\rho}_{s_1 s_2} = (\sigma_y \otimes \sigma_y) \rho_{s_1 s_2}^* (\sigma_y \otimes \sigma_y)$ and $\sigma_y$ being equal to the Pauli matrix. In terms of the coefficients of Equation (14), the concurrence results in the following:

$$C = 2|\alpha_0 \alpha_3 - \alpha_1 \alpha_2| . \tag{21}$$

With these elements, we can see now how this relation enters into the analysis of the Einstein two-slit Gedankenexperiment . To this end, we consider a variation introduced by Bohr to the Einstein setup, in which one of the two slits can move if the particle passes through while the other is fixed. The slit is attached to a spring that, if it is sufficiently sensitive, provides the which-path information (the particle-like feature) that we are looking for. Of course, if the slit is not sensitive enough, we could not know which slit the particle passed through and an interference pattern would be restored. As reported in [26], we can build the model as follows. The particle's state in the upper slit is denoted by $|u\rangle$ and that in the lower slit is denoted by $|d\rangle$. Since the slits are distinguishable, these two states are orthogonal. We can model the spring as a one-dimensional harmonic oscillator, in which $|0\rangle$ denotes the ground state and $|\alpha\rangle$ is a state corresponding to the motion of a slit with the following amplitude:

$$\alpha = \frac{\mathrm{i}p}{\sqrt{m\omega\hbar}} . \tag{22}$$

When the particle interacts with a slit, it produces a general entangled state that we can express as

$$|\psi\rangle = c_u |u\rangle |\alpha\rangle + c_d |d\rangle |0\rangle . \tag{23}$$

The particle crosses the slit in state $|\alpha\rangle$ with probability $p_u = |c_u|^2$ and the slit in state $|0\rangle$ with probability $p_d = |c_d|^2$. As in [26], the concurrence results in the following:

$$C = 2|c_u c_d| \sqrt{1 - \mathrm{e}^{-|\alpha|^2}} , \tag{24}$$

visibility is determined by:

$$V = 2|c_u c_d| \mathrm{e}^{-|\alpha|^2/2} , \tag{25}$$

and the predictability is:

$$P = \left| |c_u|^2 - |c_d|^2 \right|. \tag{26}$$

In agreement with the original Einstein–Bohr setup, we can assume that, initially, the probability for the particle to pass through the two slits is the same. This is the case in which, from Equation (26), $P$ results in null (of course, if the probability of passage between the two slits is the same, then we will obtain the maximum ignorance of the which-path information). Then, the system will be described by the following state:

$$|\psi\rangle = \frac{1}{\sqrt{2}} \left( |u\rangle |\alpha\rangle + |d\rangle |0\rangle \right). \tag{27}$$

Thus, from Equation (24), the concurrence results in the following:

$$C = \sqrt{1 - e^{-|\alpha|^2}}, \tag{28}$$

and, from Equation (25), the visibility is given by the following:

$$V = e^{-|\alpha|^2/2}. \tag{29}$$

We can now interpret these results. If we set $\alpha = 0$, then we recover the original Young double-slit experiment: the entanglement is null and the visibility is, at maximum, $V = 1$. This corresponds to a macroscopic slit, in which the particle cannot affect the motion of the first enough to be detected. In the opposite case, when $\alpha$ is very large, we obtain the complete which-path information. In this scenario, the entanglement turns out to be the maximum and the visibility vanishes. In order to have an entanglement that is sufficiently large, we can set, for example, $\alpha = 1$. This implies that the final state of the oscillator contains one vibration quantum, which is sufficient enough to be distinguished from its initial ground state. Such an analysis shows that the duality, expressed only in terms of visibility and predictability, cannot completely characterize a quantum system. There are ranges for parameters $\omega$, $m$, and $p$ for which the system manifests both the the wave-like and particle-like aspects. Moreover, from the triality relation, we can see how the system passes from a quantum regime to a classical one: by keeping, for example, $\omega$ constant and by changing the mass $m$, it is possible to span a transition from the microscopic regime to the macroscopic one. Moreover, the entanglement is related to another very important quantity: the distinguishability $D$. This quantity, which is analogous to the predictability, is a measure of the possible path information that we can obtain. It is essentially the maximum probability with which the possible $n$ paths can be distinguished unambiguously and, in a bipartite two-dimensional system—similar to that described before—results in the following [28]:

$$D^2 = P^2 + C^2. \tag{30}$$

This quantity is very useful for studying quantum systems in a multi-path setup with a path detector. As argued in [29], these aspects are at the root of a great deal of recent results in the realm of quantum optics. In addition, as stressed in [30], the consistency between the experimental data and Equation (11) suggests that the entanglement requires a revision of quantum duality, thus modifying this relationship into quantum triality.

## 3. Entanglement and Modern Physics Teaching

The aim of this section is to explore the concept of quantum entanglement in order to evaluate the most effective ways through which to present the gist of it to high school students.

As anticipated in the introduction, in recent years, we have witnessed what the experts call the Second Quantum Revolution [10]. While this revolution has not been brought forward by new discoveries in underlying physical theory—non-relativistic QM in its

established form is nearly a century old—it differs from the "first" quantum revolution (i.e., the technological innovations arising from the application of the new quantum discoveries in the years following the Second World War) because, while the first revolution was based on the behaviors of many-particle quantum systems (e.g., semiconductors, lasers, and nuclear devices), the second one is based on the manipulation of single quantum particle states. To promote and advance these new quantum technologies in the European Union, in May 2016, a Quantum Manifesto was published [31]. Among its aims, the necessity of providing both educational programs for scientists and technicians, as well as information for the public at large, was underlined. The prerequisite for the successful attainment of these two goals was the development of an effective strategy to allow high school students "to grasp the essence, potential and social implications of new quantum technologies" [32].

Paradoxically, while in higher education, "classical skills in physics and engineering are valued just as much, if not more than, knowledge of quantum information science, for the majority of roles currently in the quantum industry" because "of the focus on the development of hardware, and the fact that quantum information theory lives in its own abstract space independent of the hardware" [33]. As such, in view of the aforementioned goals stated in the Quantum Manifesto, the role of modern physics in high school curricula cannot be emphasized enough because we are talking about the conceptual basis on which the whole structure of the quantum revolution rests. Now, the peculiar difficulties of teaching quantum physics in high school are well known and, in our opinion, they are mainly caused by two factors: the sophistication of the mathematical tools required by QM and the absolute lack of consensus among scientists about what the correct interpretation of QM should be.

The difficulty of the task will not deter us from trying. Before devising a strategy to start teaching the basics of QM in high school, it is expedient to identify the fundamental tenets of the discipline. In this regard, we echo the words of Erwin Schrödinger himself, who, in referring to quantum entanglement, wrote the following: "I would not call that one but rather the characteristic trait of quantum mechanics, the one that enforces its entire departure from classical lines of thought" [34]. Not only is it entanglement that, in the end, differentiates a quantum system from a classical one, but this very qualitative difference is what makes possible the quantum information technology that is paramount in the Second Quantum Revolution. This explains the rationale behind the aims that we stated in the first part of this section.

*3.1. The EPR Paradox*

The aforementioned quote by Schrödinger about entanglement was from a paper that he wrote in response to the famous work by Einstein, Podolsky, and Rosen about the supposed incompleteness of QM [6] that we extensively discussed in Section 2. Here, we will not use the original account contained in the EPR paper but a simpler version by David Bohm, which has been used in many recent works dealing with quantum foundations (see, for example, [35]). Let us consider a system composed of two electrons, each of which can be in two different spin states relative to the *z*-direction, which is spin up $|\uparrow\rangle$ and spin down $|\downarrow\rangle$ i.e. the eigenstates of the operator of *z* component of the spin: $\hat{S}_z|\uparrow\rangle = +\frac{\hbar}{2}|\uparrow\rangle$, $\hat{S}_z|\downarrow\rangle = -\frac{\hbar}{2}|\downarrow\rangle$. The Hilbert space that describes this two-particle system contains all the vectors that can be obtained by a linear combination of a tensor product of any state that describes a single particle state. Among all of these states, we will focus our attention on the following state:

$$|\psi_{\text{EPR}}\rangle = \frac{1}{\sqrt{2}}(|\uparrow\downarrow\rangle - |\downarrow\uparrow\rangle), \tag{31}$$

where $|\uparrow\downarrow\rangle$ is a shorthand for $|\uparrow\rangle \otimes |\downarrow\rangle$. QM states that, if we prepare our two electrons in the state shown in Equation (31), we will pull them apart and let two experimenters measure their spins, and each of them will find the measured electron in the spin up or spin down state with a 50% probability. The two measurements will not be uncorrelated: for example, if the first experimenter finds the first electron to be spin up and the state

of the system collapses to $|\uparrow\downarrow\rangle$, then the second experimenter will invariably find the second electron to be spin down. The argument in the EPR paper is that this 'instantaneous collapse of the wavefunction' is some sort of spooky action at distance, i.e., a concept that was already suspect when Newton introduced it when described gravitational attraction but is now simply untenable in the light of special relativity. Therefore, if this scenario is not possible, the observed correlations must be accounted for by another factor present in the system from the outset. However, since the mathematical framework of QM does not include this information within the $|\psi_{\mathrm{EPR}}\rangle$ state, it suggests that QM itself is incomplete. In other words, there are physical properties of the system that are not represented by quantities in the theory. The objections of the EPR paradox to the current formulation of non-relativistic QM are equivalent to the assumption that probability can only be classical probability, i.e., a measure of our ignorance of the system and not one of its ontological features. This basically means that it is possible, in principle, to devise a purely classical model of quantum systems that deterministically encodes all the possible results of future measurements via the additional—when compared to the allegedly incomplete current quantum theory—quantities dubbed hidden variables [36]. The underlying concept here is that, during a quantum measurement, there exist countless unknown variables within both the system and the measuring apparatus that cannot be precisely monitored. Nevertheless, these variables have the capability to impact the outcome of the measurement, thus leading to seemingly random results [37]. In the end, as we anticipated in the previous section, about thirty years after the original EPR paper, John Stuart Bell—a physicist from Northern Ireland—managed to prove [7] that any (local) hidden variable theory is incompatible with the prediction of QM.

Thus, what makes the story of the EPR paradox so important in the history of QM is that it arises only owing to the existence of entangled states and that its 'resolution' is connected to demonstrating the impossibility of local hidden variables. We already discussed some of the theoretical details of these two aspects in Section 2; at this point, we will present in the following subsections, some ideas on how to introduce, respectively, the basics of entangled states and Bell's inequalities to high school students.

*3.2. Transitioning between Classical and Quantum Paradigms: From Classical Correlations to Entangled States*

3.2.1. Introduction: Ideas to Introduce a New Paradigm

It is important, in teaching QM and before introducing the concept of entanglement, to suggest to students some points for reflection, i.e., concepts to be analyzed in a targeted and coherent manner with the final purpose of the teaching action. It is crucial to furnish students with a comprehensive framework of QM, to acquaint them with its epistemological origins as recounted by history (read Section 2 in this regard), and, ultimately, to underscore and address the most prevalent questions that arise in the minds of those endeavoring to comprehend the quantum world. As we have extensively discussed in the previous sections when dealing with a quantum system, where the laws of classical physics can no longer be used to describe its state, practically everything we know about a classical physical system no longer holds true, and, instead, there are different laws at play—the laws of QM.

But what does it mean for quantum systems to exhibit different laws?

Should we then divide the universe into two subsets labeled "classical" and "quantum"?

The answer is no, there are no "classical" systems and "quantum" systems in our unique universe.

The physical laws governing the universe are the same everywhere. However, it is important to consider that, at microscopic scales, particle velocities are extremely higher than those we are accustomed to in our daily lives, i.e., their masses are extremely small when compared to those in the macroscopic world. This implies the emergence of entirely different phenomena when transitioning from one scale of magnitude to another [38]. Newton's laws of classical mechanics and Maxwell's laws for electromagnetic fields are capable

of describing, with a good approximation, the phenomena that occur for macroscopic objects moving at not too high speeds, just as QM explains what happens on atomic and subatomic scales. Quite simply, for us, who are too large and possess senses too slow to detect quantum effects, the reality we perceive can be accurately described by Newtonian physics. Thus, we can assert that classical physics provides a satisfactory approximation of quantum physics. This happens because the quantum effects in our everyday reality are negligible, and classical physics can effectively simulate and predict the macroscopic reality without taking quantum effects into account [39]. On the other hand, if we want to study microscopic systems, systems at a quantum scale, such as the motion of a molecule or the characteristics of an electron, quantum effects are no longer negligible, and Newton's laws and Maxwell's equations are no longer sufficient. Our senses are not adequate and are not designed to perceive and visualize these phenomena: this is why everything at this level seems so incomprehensible to us.

How, then, do we explain experimental results? Through an appropriate abstract analysis based on mathematics. Therefore, a more or less in-depth understanding of the abstract mathematical model that can describe the microscopic system is necessary. The mathematics of QM is not overly complicated; it is primarily based on the concepts of vector spaces, and, thanks to the symbolic notations introduced by theorists (i.e., bra-ket notation, also known as Dirac notation or Dirac formalism) over time, it has become quite manageable. What we aim to demonstrate is that it is possible to comprehend quantum phenomena using an even simpler mathematics, a purely symbolic mathematics that will lead us to uncover where the most peculiar physical properties of quantum systems arise, which have also been experimentally verified.

However, we need to establish some key points in our discussion with the students:

1. If you know the state of a quantum system, you do not know everything there is to know about the system. In particular, it is not guaranteed that you can predict the outcome of an experiment. The state space of a quantum system is not a countable set, and, in a quantum system, states are not distinguishable from each other in a completely unambiguous way.

2. Moreover, in a quantum system, it is not possible to perform an experiment (a measurement) that leaves the system undisturbed, regardless of how gentle the measurement itself might be. In essence, a quantum system, whether it is an electron, a photon, or a collection of atoms, does not have a well-defined state; more precisely, it exists in a condition of a superposition of states (in this regard, the "Schrödinger's Cat Paradox" illustrates this concept well), which are all equally realizable in terms of stochastic probability. It is only through the measurement of a specific physical quantity associated with the system that it collapses into one of the possible states.

3. These effects can be observed in quantum systems that are composed of a single particle, but they are not the only distinctions we can observe between systems composed of classical objects and quantum objects. There are additional differences that manifest in composite quantum systems that include at least two subsystems, each of a quantum nature. The correlations between these subsystems give rise to another distinction between classical and quantum systems as, while correlations in classical systems can always be described in terms of classical probabilities, this is not always possible in quantum composed systems. Such non-classical correlations lead to apparent paradoxes, like the famous EPR Paradox, which might suggest, at first glance, the existence of a remote and non-local action in QM. States that exhibit such non-classical correlations are referred to as entangled states.

The purpose of this chapter is to introduce the basic tools that allow us to understand the nature of such states, i.e., to distinguish them from classically correlated ones and to quantify, as much as possible within our simplified analysis of the phenomenon, the non-classical correlations. In order to explain as simply as possible what happens in an entangled state and to make the understanding of the phenomenon easier for a reader whose knowledge is at the level of a high school student, we will modify and simplify the

mathematical notation commonly used in QM. We will rely on the reader's familiarity with the notions of probability of independent and incompatible events, linear combinations of two vectors, and mathematical functions. Analogies will often be used solely to facilitate the understanding of the main features of the physical properties found in the various systems analyzed. It should be emphasized that the purpose of this treatment, as seen from the use of non-mathematical symbols, is to highlight the physical characteristics of an entangled system in a qualitative rather than quantitative manner while focusing on the mathematical origin of the physical phenomenon. Therefore, it will be necessary to restrict the analysis to bipartite quantum systems, that is, systems composed of only two subsystems.

### 3.2.2. Pure States

At the outset, let us consider a system composed of a single electron in a pure state, which is denoted as $\Psi_e$. We can identify a pure state as a vector that contains information about the probabilities of the possible outcomes of a measurement. These probabilities are associated with a system that is appropriately "prepared" in this state, which is then referred to as a pure state. Suppose we want to measure the spin of this electron along the $z$-axis of our reference system. The electron's state in this regard, referred to as a quantum superposition, before measurement remains undefined, as shown in Figure 2. However, following the measurement, it will "transform" into either $|\uparrow\rangle$ or $|\downarrow\rangle$.

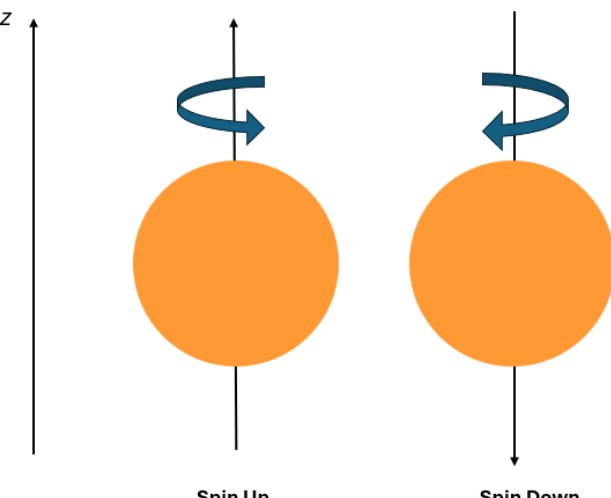

**Figure 2.** Quantum superposition of the intrinsic angular momentum of the electron along the $z$-axis.

The two arrows indicate whether the electron's spin is aligned or anti-aligned with the orientation of the $z$-axis. The superposition state can be characterized by a probability function that contains the probability of obtaining one of the two outcomes ($\downarrow$, $\uparrow$) as follows:

$$|\Psi_e\rangle = \Psi_{up}|\uparrow\rangle + \Psi_{down}|\downarrow\rangle. \tag{32}$$

In a completely analogous manner, one could write a probability function to study the outcome of flipping a coin, where the possible "measurement" results are "Head = **H**" or "Tail = **T**", as shown in Figure 3. One can express an analogy of this via Equation (33):

$$|\Psi_{coin}\rangle = \Psi_{head}|H\rangle + \Psi_{tail}|T\rangle \tag{33}$$

It is evident that, even from the analogy with the coin, the probability of obtaining $|\uparrow\rangle$ or $|\downarrow\rangle$ from the measurement is equal to $\frac{1}{2}$, and the two events are mutually exclusive; the two events are mutually exclusive (incompatible) when the occurrence of one event excludes the occurrence of the other and the sum of their probabilities is equal to 1. In QM,

the probability of a single event is represented by $(\Psi_{\text{up}})^2$ or $(\Psi_{\text{down}})^2$; hence, the values of $\Psi_{\text{up}}$ and $\Psi_{\text{down}}$ in Equation (32) are $\frac{1}{\sqrt{2}}$ .

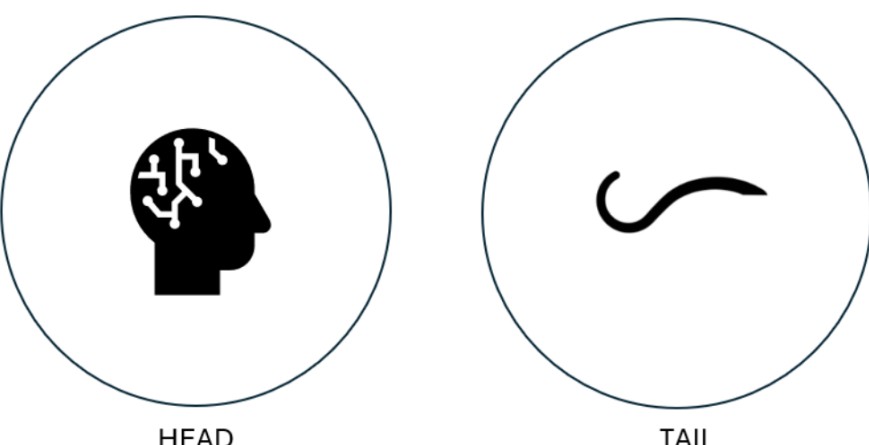

**Figure 3.** The outcomes achievable from tossing a coin.

### 3.2.3. Bipartite Quantum Systems

The analyses performed so far have been carried out on systems composed of a single element or "quantum state," which we referred to as $\Psi_{\text{e}}$ and by analogy $\Psi_{\text{coin}}$. At this point, let us consider a bipartite quantum system, i.e., a system composed of two pure quantum states on which we intend to perform measurements on the individual subsystems that make up the overall system. For instance, we could consider a system composed of two electrons in the pure state $\Psi_{\text{e}}$, which we shall label as $\Psi_{1\text{e}}$ and $\Psi_{2\text{e}}$, respectively. The state of the overall system can be identified as $\Psi_{\text{system}}$. Now, imagine that we can perform a measurement on each electron separately and obtain a completely random result, which is denoted as *a*; as such, we will take different values each time, which we will record. Later, we perform a measurement on Subsystem 2 and record the outcome. Upon comparing the measurements on electron-1 and electron-2, we observe that they are completely independent of each other. The same thing happens with a system composed of the two coins: $\Psi_{1\text{coin}}$ and $\Psi_{2\text{coin}}$. When we inquire about the probability of obtaining a certain configuration among the various possibilities upon flipping them, we will then summarize all the possible configurations, as shown in Table 1.

**Table 1.** Possible configurations obtainable from the toss of two coins.

| Coin 1 | Coin 2 |
|:---:|:---:|
| $\lvert H \rangle$ | $\lvert T \rangle$ |
| $\lvert T \rangle$ | $\lvert H \rangle$ |
| $\lvert H \rangle$ | $\lvert H \rangle$ |
| $\lvert T \rangle$ | $\lvert T \rangle$ |

Suppose we want to calculate the probability of obtaining the first configuration shown in Table 1 after flipping. We need to consider that what happens to the first coin will not influence the second coin in any way. In other words, the two events can be considered independent. Two events are said to be stochastically independent when the occurrence of one event does not change the probability of the other event occurring. For the first coin, the probability of obtaining $\lvert \mathbf{H} \rangle$ is $\frac{1}{2}$, and, similarly, for the second coin, the probability of obtaining $\lvert \mathbf{T} \rangle$ is $\frac{1}{2}$. Consequently, the probability of having the first configuration is given by the theorem of compound probabilities for independent events. That is to say that the (composite) probability of the occurrence of two independent events is simultaneously equal to the product of the probabilities of the individual events: $P(A \cap B) = P(A) \cdot P(B)$, so $P(\lvert \mathbf{H} \rangle / \lvert \mathbf{T} \rangle) = P(\lvert \mathbf{H} \rangle) \cdot P(\lvert \mathbf{T} \rangle)$ is equal to $\frac{1}{4}$. The four configurations are all equally

likely, and each one is associated with a probability of $\frac{1}{4}$. To provide another example, let us consider two dices and aim to calculate the probability of obtaining a 6 on the first die and a number between 1 and 5 on the second die, as shown in Table 2. Let us recall that the probability of an event is defined as the ratio between the number of favorable outcomes, which lead to the occurrence of the event, and the total number of possible outcomes. In this case, the probability will be the following:

$$P = \frac{1}{6} \times \frac{5}{6} = \frac{5}{36}. \tag{34}$$

**Table 2.** Probability of a bipartite system of two dice delivering a 6 on the first die and a number between 1 and 5 on the second die.

| First Die | Second Die |
|:---:|:---:|
| 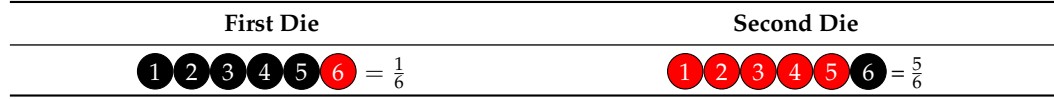 | |

In Equation (34), just as in the case of coins, it is possible to uniquely identify which of the two factors in the product represents the "probability component relative to the first subsystem"$= \frac{1}{6}$, and the "probability component relative to the second subsystem" $= \frac{5}{6}$. The state of such a composed quantum system, in the case of the two electrons, can be represented by the following Equation (35):

$$|\Psi_{\text{system}}\rangle = |\Psi_{1e}\rangle \otimes |\Psi_{2e}\rangle. \tag{35}$$

This arises from the product of the relative probabilities of the first and the second subsystem. As a quantum physicist would say: the system's state is factorizable into the states of the two subsystems. This mathematical property precisely contains the characteristic of independence between the states, as can also be seen in Figure 4, i.e., the measurement on the individual subsystem has the effect of yielding the value *a* for the considered subsystem and does not alter the state of the other, thus acting on the latter as an identity operator.

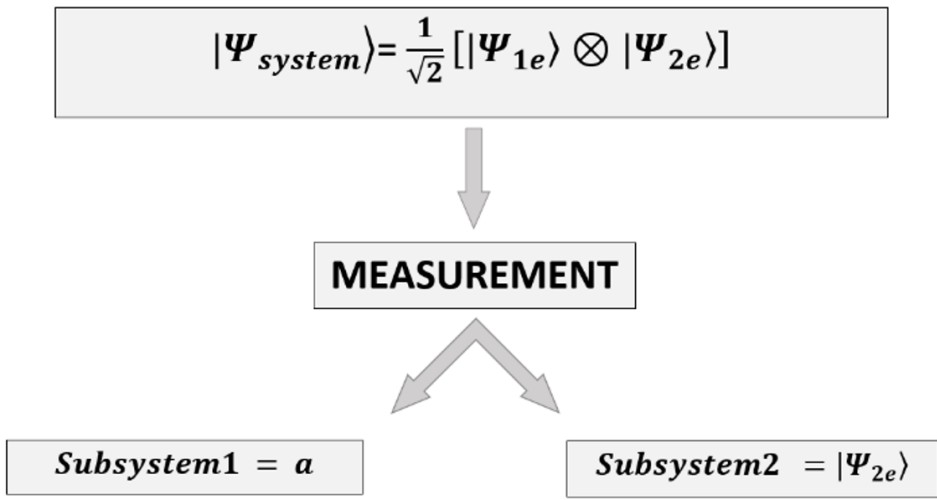

**Figure 4.** The effect of the measurement on the two subsystems. Particularly, on *Subsystem*1, we have the value *a* for *Subsystem*1, and *Subsystem*2 substantially remains unaltered in a indeterminate state until we measure a property.

3.2.4. Entangled System

The pure states previously defined, of which we have seen the main physical characteristics, can be linearly combined using the principle of superposition to yield a new pure state. Its mathematical representation is as follows:

$$|\Psi_{\text{system}}\rangle = \frac{1}{\sqrt{2}}(|\Psi_1\rangle \otimes |\Psi_2\rangle + |\Phi_1\rangle \otimes |\Phi_2\rangle), \tag{36}$$

where $|\Psi_1\rangle \neq |\Phi_i\rangle$ and ($i = 1, 2$). Through observing Equation (36), we can first highlight that the "+" operator indicates two addends on the right-hand side, which represent the states of the two subsystems comprising the global system as follows:

$$\text{Subsystem } 1 = \frac{1}{\sqrt{2}}(|\Psi_1\rangle \otimes |\Psi_2\rangle), \tag{37}$$

$$\text{Subsystem } 2 = \frac{1}{\sqrt{2}}(|\Phi_1\rangle \otimes |\Phi_2\rangle). \tag{38}$$

We thus have a composite system formed by two subsystems, each with the characteristics analyzed earlier. To understand what happens in this bipartite system, we must consider that we are constructing a new system that applies the superposition principle with the use of the "+" operator, thus adding the intrinsic features of the two constituent subsystems. The composite operation present in Equation (36) is entirely permissible from a mathematical perspective. We know that it is always possible to carry out the linear combination of vectors in a vector space; furthermore, this is achievable in a laboratory setting. In essence, the state $|\Psi_{\text{system}}\rangle$ from Equation (36) has a mathematical significance and a real physical existence [40]. We wonder what happens if we now proceed to perform the usual measurements on the two subsystems independently. Observations tell us clearly that the states of the two subsystems are not independent. It has been observed that the measurements carried out on one of the two subsystems influence the state of the other.

Let us first try to clarify what happens from a mathematical standpoint through the following step-by-step analysis:

1.  First, we will associate the symbolic operation "•" with the complex of operations formed by the operators $\otimes$ and $+$, in which "∘" represents the appropriate wave functions $|\Psi_i\rangle$ and $|\Phi_i\rangle$, as shown in Equations (36) and (39):

$$\bullet = [\circ \otimes \circ + \circ \otimes \circ]. \tag{39}$$

2.  Let us associate an amount of information, which is denoted as $\triangle_1$ and $\triangle_2$, with the physical state of the two subsystems. The so-defined composite operation "•" will provide us with the measurement result $a_1$ for Subsystem 1 and the value $a_2$ for Subsystem 2 in the following manner:

$$a_1 = \bullet(\triangle_2), \tag{40}$$

$$a_2 = \bullet(\triangle_1) \tag{41}$$

3.  We observe that the measurement $a_1$ on the first subsystem will be a function of the information contained in the second $\triangle_2$, and vice versa.

The results are shown in Table 3.

**Table 3.** First line: the information inside the subsystems. Second line: the measurement values. Third line: the measurement value calculation.

| Subsystem 1 | Subsystem 2 |
|:---:|:---:|
| $\triangle_1$ | $\triangle_2$ |
| $a_1$ | $a_2$ |
| $a_1 = \bullet(\triangle_2)$ | $a_2 = \bullet(\triangle_1)$ |

The two systems are not independent; the state of one depends on the information contained in the state of the other, as shown in Figure 5. It is evident that the cause can be attributed to the nature of the linear combination operation embedded in the operators $\otimes$ and $+$, which results in this "insoluble entanglement". The state of the composed system cannot be factorized into a product of independent states, and such a system is termed an entangled system.

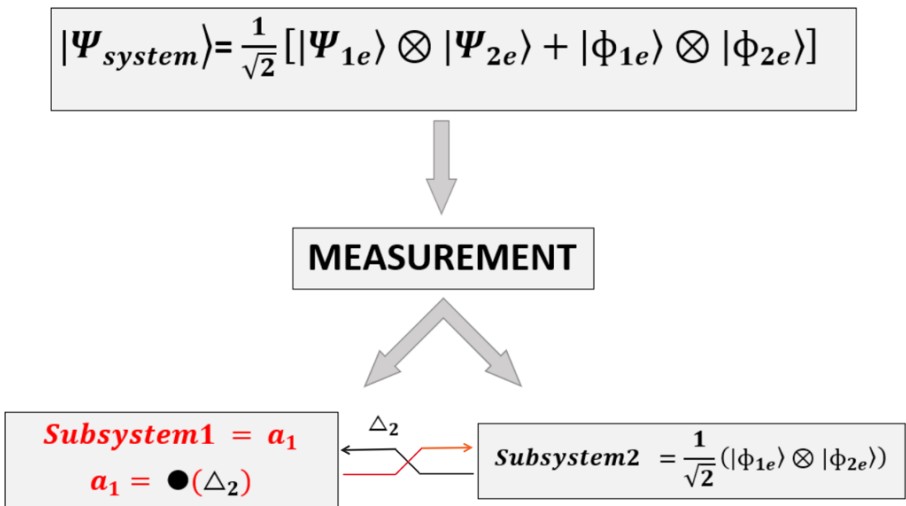

**Figure 5.** A conceptual diagram for a measurement on an entangled system. Particularly, on *Subsystem*1, we have the result of the measurement, which is a function of the information $\triangle_2$ of *Subsystem*2. The color and direction of the arrows indicate the hypothetical origin and hypothetical direction of the flow of information (red for Subsystem1 and black for Subsystem2). The *actual origin* of the arrows from a specific subsystem and the *actual belonging* of the information to a specific subsystem is not verified. The scientific debate on this topic is still completely open and currently represents the true mystery of entanglement. The image presented along with the accompanying captions aims to be an educational aid aimed at understanding the complexity of the phenomenon.

So, we can resume our experiment in the following steps:

1. Preparation: The entangled system is prepared in a particular state, such as $|\Psi_{\text{system}}\rangle$.
2. Measurement Setup: Separate measurements are performed on each subsystem, which is indicated by the operators $\otimes$, $+$, and "$\bullet$" in the context of QM.
3. Quantum Interaction: The measurement on one subsystem influences the other, thereby causing a change in their states due to entanglement.
4. Outcome: After the measurements, the specific values, $a_1$ and $a_2$, are obtained for each subsystem.
5. Correlations: The outcomes for each subsystem are correlated in such a way that they cannot be described independently; their behavior is intertwined.
6. Entanglement Effect: The measurements on one subsystem provide information about the other, thereby defying classical concepts of independent measurements.

What are the physical implications of such a "mathematical entanglement"?

A measurement on Subsystem 1 therefore requires information that is indefinitely contained (i.e., not yet actualized through a measurement) in Subsystem 2. In other words, we are saying that, to obtain the measurement result $a_1$, we need to retrieve information from Subsystem 2, which—according to QM—has not yet reached a definite state and therefore does not possess definite information to "communicate". Let us hypothesize, however, that such information exists, i.e., that it is already recorded in detail (without a probabilistic cloud) in Subsystem 2, thus contradicting the principle of superposition and asserting that Subsystem 2 is in a condition of a defined state. Further, let us assume that this information is somehow transferred to Subsystem 1. These pieces of information would need to travel through a "physical channel" at a speed that is, at most, the speed of light. Well, experimentally, it has been observed if we separate the two subsystems in an adequate manner that is posing them at a distance such that light cannot travel between them in the time interval between the two measurements on the two subsystems are made, then the exchange of information we have referred to as $\triangle_1$ and $\triangle_2$ still occurs [41]. In other words, if we perform a measurement that involves a spatial separation, then the correlation between entangled particles seems to violate the principle that no information or influence can travel faster than the speed of light. This phenomenon is at the heart of the EPR paradox and highlights the non-local nature of entanglement in quantum systems. It challenges our classical intuitions about causality and the limitations of information transfer, thereby showcasing the unique and puzzling nature of quantum interactions. Hence, we need to hypothesize the presence of actions at a distance, whose speed is superluminal (another contradiction!). This characteristic of entangled systems is known as non-locality, i.e., the actions that manifest between two entangled systems do not have a local nature; in other words, they do not depend on the relative distance between the two systems [42]. To make sense of this, one might consider that QM is not a complete theory—essentially, its way of representing microscopic phenomena might lack certain variables. This implies that not everything that needs to be taken into account for microscopic systems has been considered. Therefore, hidden variables might exist that are capable of explaining these non-local effects and thus making the quantum system deterministic (thereby contradicting the principle of superposition and deeming QM an incomplete theory!). The contradictions just highlighted, i.e., the non locality of quantum actions and the hypotheses of incompleteness in QM, have confounded physicists for about thirty years until the physicist John Bell demonstrated that the distinctive features of entangled systems are intrinsic qualities of quantum systems. He showed that QM, as formulated, is a complete theory.

### 3.2.5. Summary

In conclusion, the physical repercussions of mathematical entanglement, as described by Equation (36), are profound and astonishing. They demonstrate that, at the quantum level, entangled particles are interconnected in ways that cannot be explained by the laws of classical physics. Some of the most significant repercussions are as follows:

(a)   Instantaneous Correlations: Even when entangled particles are separated by large distances, any measurement made on one particle will instantaneously influence the state of the other, regardless of the distance. This phenomenon appears to violate the concept of causality in classical physics.

(b)   Quantum Communication: Quantum entanglement can be harnessed for quantum communication, such as in the field of quantum cryptography. Changes to one entangled particle can be detected instantaneously by the other, thus allowing for the transmission of secure information.

(c)   Quantum Computing: Entanglement offers significant advantages in the field of quantum computing. Entangled qubits, which are qubits that are part of an entangled quantum system, can exist in combined states and perform complex operations in parallel, thereby potentially speeding up the solution of problems that are otherwise impossible for classical computers.

(d) Representation of Quantum Reality: Entanglement demonstrates that the laws of QM can lead to results that seem counterintuitive or contrary to our everyday experience. This underscores the need to embrace a new conceptual paradigm when describing the world at the quantum level.

(e) Quantum Thermal Engines: Theoretical concepts of quantum engines, though not yet practically implemented, leverage entanglement to explore innovative ways of enhancing efficiency in converting heat to work, thus challenging traditional conceptions of physical reality and paving the way for potential applications in the fields of thermodynamics and engineering [43].

In summary, the mathematical entanglement in quantum systems leads to phenomena that challenge traditional conceptions of physical reality, thereby paving the way for innovative applications in a broad range of fields spanning from communication to computation to engineering sciences, as well as in the understanding of the fundamental nature of matter and the universe.

### 3.3. Transitioning between Classical and Quantum Paradigms: Bell's Inequality with Scratchcards

We shall now illustrate the essence of Bell's results, which was anticipated in Section 2.3, in a manner that will be also understandable to high school students. We shall be using a Gedankenexperiment that was firstly devised by David Mermin [44], which features Stern–Gerlach-like measurements on the entangled state in Equation (31). Our goal is to devise a classroom activity that allows students to grapple with the practical impossibility of having a hidden variable theory that is able to predict the actual measurement results of QM. This activity involves the use of fictional scratchcards—a scratchcard, also known as scratch-it or scratch-and-win, is a small card or ticket with a concealed area that can be scratched off to reveal a hidden message, image, or code, and it is often used for games of chance, promotions, or prizes. In our case, these fictional scratchcards are used to represent the supposed "complete" state that, according to Einstein, Podolsky, and Rosen, should actually describe a quantum system. We obtained the idea of using scratchcards from a popular science book by physicist Colin Bruce [45], although we also completely changed their structure and use. Here, we detail the proposed classroom activity describing the preliminary information (premise) and the specific task (problem) the students will be given.

Premise: a mysterious, recently established company distributed a very peculiar set of scratchcards for a raffle. Their outward appearance is shown in Figure 6.

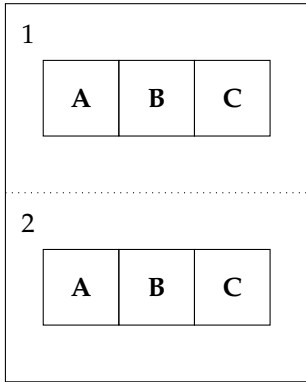

**Figure 6.** Scratchcard with two halves, perforation, and labeled squares.

The scratchcard is divided into two halves—numbered, respectively, 1 and 2—with a perforated line on the middle that allows one half to be ripped off from the other. On each half, there are three squares—labeled A, B, and C, respectively—which are covered by an opaque substance that can be scratched to reveal the content of the square beneath. Each square can be either white or black, and its color can only be revealed by scratching its cover. Only a single square per half can be scratched to reveal the color beneath;

if one tries to scratch more than one square, the card simply flashes into flames, as if there is a clever chemical self-destruction device devised to prevent this from happening. A conspicuous amount of these scratchcards was collected and examined, where one square per half was scratched and the results were recorded. After comparing the results from the corresponding halves, the following correlations were found:

- If one scratches the corresponding square (i.e., with the same identifying letter, A, B, or C) on both halves, then the color revealed beneath is always the same. It is white 50% of the times and black the remaining 50% of the times.
- If one scratches different squares (i.e., squares identified by different letters) on two halves from the same ticket, one finds the same color only 25% of the time (with an equal probability for each color appearing) and different colors 75% of the time (with an equal probability of finding black on the half labeled 1 and white on the half labeled 2, and vice versa).

This makes the lottery a fair game, as the rules written behind each scratchcard say that a scratchcard costs EUR 1.00, and if players find squares of the same color in non-corresponding squares of the two halves of the same card, they will win EUR 4.00.

Problem: You are an employee of a public company that has the task of preventing fraud in lottery games. In actuality, a fraud per se is not suspected because there is plenty of empirical evidence that the probabilities of obtaining the winning results are those described in the preceding paragraph, but since the mysterious company that flooded the market with the aforementioned scratchcards has not yet revealed how they are made and since the self-destruct mechanism prevent direct inspection, your boss thinks that something is amiss. Therefore, your company has tasked you with discovering how the tickets could have been created in order to match the empirically observed frequencies.

The task, as presented, seems to have no connection with QM, and this is deliberate. The students will approach it as a type of mathematical puzzle and will ponder: how can the scratchcards be created? What follows is a potential approach to solve this enigma. We do not assert that it is the sole possible approach, but it should suffice to illustrate the utility of the suggested activity.

The first evidence to be explained is the fact that, if corresponding squares are scratched on the two halves of every given card, the color revealed beneath is the same. This implies that the two halves must share an identical sequence of colors in their A, B, and C squares. If we suppose that, for the sake of argument, they do not match, we must conclude that the two sets of colors hidden under the scratch-off covers are different for at least one pair of colors. However, if someone with that card were to erase the boxes where the match is not verified, a frequency of zero for events showing different colors in corresponding boxes would not be obtained, which is contrary to empirical evidence. To be fair, one might argue that there might actually be different halves, and that it simply never happens that someone scratches them, thereby revealing the unmatched pair. This objection, although unlikely, cannot be completed ruled out. See below.

The second evidence that requires explanation is the fact that, if one scratches different squares on two halves from the same ticket, one finds the same color only 25% of the time and different colors the remaining 75% of the time. Once we have established that the two color triplets on any given scratchcard must be identical, it becomes clear that one cannot use triplets like black, black, and black or white, white, and white, if the possibility of finding different colors in different squares in the two halves is to be accounted for. With three possible positions for only two colors—considering we have just ruled out triplets with the same color repeated three times—the only viable triplets will involve one specific color (black or white) in one square (A, B, or C), and the opposite color (white or black) in the other two squares. Therefore, if we pick, at random, one square in the 1st half and another non-corresponding square in the 2nd half of the card, the probability of obtaining the same color in both squares is 2/3. But, this is different from the value of 25% (i.e., 1/4) that we obtained from the observed frequencies! We might suppose that the problem lies in the exclusion of the triplets with the same color in all the squares but, even if we

put them back them in the mix in any proportion, the average probability of finding the same color will increase. Thus, this will not decrease and will exacerbate, rather than solve, the problem.

We have thus arrived at an unanticipated conclusion. There is no way to devise a scheme of colors under the scratch-off boxes that will reproduce the observed results. A possible justification for the objection stated two paragraphs above is assuming that there is something that actually prevents us to freely choose at random the squares to be scratched. In a display of some out-of-the-box thinking, one might also argue that, if the makers of the scratchcards were so ingenious to have them self-destruct when trying to scratch more than one square per half, they could also install on them some sort of communication device that transmit information between the two halves and have them change color accordingly. But more on these objections later.

At this point, we can elucidate on the connections between all of these analogs and QM. These 'impossible' scratchcards actually exist: the two halves of the same card represent the two entangled electrons in the singlet state of Equation (31). Choosing one of the three boxes with the letters A, B, and C simulates the experimenter selecting, on a Stern–Gerlach-like apparatus, whether measuring the particle's spin in the vertical direction or in a direction that is $\pm 120^o$ off, as well as the scratching of the corresponding box, which simulates the spin measurement itself and its result. The frequencies provided, in relation to the scratchcards, mirror those that would be obtained in actual spin measurements using the aforementioned setup.

The idea of assuming there is a prearranged scheme of colors under the scratch boxes is equivalent to the assumption that hidden variables exist and that they explain the seemingly paradoxical outcomes of quantum measurements. The impossibility of actually devising a scheme of colors on the scratchcards that aligns with the empirical results is equivalent to the confutation of the hypothesis of hidden variables. The fact that the probability of obtaining a given result, if the colors on the cards are prearranged in any conceivable scheme, is different from the probability of actually obtaining it experimentally is equivalent to the proof that QM does not satisfy Bell's inequality. What about the quantum equivalent of the communication device between the two halves of the scratchcard? The possibility of an ingenious mechanism to allow communication between the two halves of a scratchcard is invalidated by the requirement of the locality imposed by special relativity (where the measurement results for entangled particles are the same even for space-like separated events). The idea that the experimenter is not actually free to randomly choose the box to be scratched (or the orientation of the Stern–Gerlach apparatus), is equivalent to advocating superdeterminism, i.e., the idea that the distribution of hidden variables is not independent of the measurement settings. While superdeterminism is not ruled out per se, it is generally not considered viable by the majority of scientists, although some of them [46] still pursue it.)

### 3.4. Summary

Niels Bohr once stated the following: "those who are not shocked when they first come across quantum theory cannot possibly have understood it" (as quoted in [47]). Richard Feynman was also known to have said the following: "nobody understands quantum mechanics" [48]. These two statements, as contradictory as they may appear on the surface, are actually in perfect agreement. Bohr used the term understanding in the sense of knowing the formal mathematical framework of QM, Feynman used the same term in the sense of having an intuitive representation of it. Our proposed activity has the purpose of allowing students to experience the "shock" Bohr was referring to in realizing that QM is not conceivable in classical everyday terms, as well as also letting them realize that, with Feynman, this inconceivability of QM does not stem from their inability to devise a sound solution, rather it is an intrinsic trait of the theory itself.

Certainly, we are not suggesting this approach merely to evoke a sense of wonder among laypeople. Bohr's quote implies that recognizing that QM is not visualizable is a prerequisite for comprehending its formal framework. Thus, for someone seeking to learn QM, a clear grasp of this concept is indispensable. Its indispensability lies in the fact that such a comprehension prevents misconceptions. If we start introducing QM in high school by presenting students with a visually understandable model of entanglement (e.g., similar to the one outlined in [49]) we might be showing them some characteristics of quantum systems, but there is a risk of denying them access to the fundamental conceptual aspect we were discussing earlier. It is not possible to have a classical mechanism that describes entanglement because that would be equivalent to a hidden variables model, which we know to be impossible. On the other hand, once this point is clear, one could go on learning some of the aspects of the formalism of quantum theory (see, for example, [50]) as we are well aware that there is no substitute for it (i.e., no intuitive model of the supposed inner workings of the quantum world will suffice).

## 4. New Avenues for Quantum Technologies: Quantum Molecular Materials

Contemporary society has been impacted by novel technological fields that are based on QM principles. Indeed, currently, the possibility of addressing, controlling, and detecting individual quantum systems has led to the Second Quantum Revolution, in which QM has been exploited for the development of a new generation of technologies that have a significant potential to revolutionize several fields. This revolution is marked by advancements in quantum computing, quantum communication, quantum sensing, quantum cryptography, and quantum metrology. The involved Quantum Technologies (QTs) harness the fundamental properties of quantum systems, such as the discreteness of energy levels, superposition, entanglement, and quantum coherence, to achieve novel capabilities beyond classical limits [51]. A building block of QTs is quantum computing, which is fundamentally based on quantum bits (qubits). Quantum computing exploits the principles of the superposition and parallelism of operations not only to speed up current computation, but also to solve otherwise unsolved problems. Qubits can represent multiple states simultaneously, thus allowing for complex calculations; this fact is very revolutionary with respect to the only two logical states in classical computation [52]. Several physical systems can be used as qubits, but they must address the following requirements to be suitable as well-working qubits. They must have long coherence time and be characterized by an easy initialization to a specific state. They also must be scalable structures that can address the miniaturization process of electronics, and they must also be individually measurable (even if they are used in complex quantum gates) [53]. Photons [54], cold atoms, impurities in solids [55], superconducting devices [56], and many other physical structures have been investigated as qubit candidates, see Figure 7. The most advanced qubits are based on superconductors, especially when using Josephson junctions. Indeed, these are used in quantum computers that were developed by the major companies like IBM, Google, and Microsoft. The main limit of this technology is the cryogenic operating temperature together with the difficulty to integrate a large number of qubits. A good alternative is constituted by the use of spin since it is an intrinsically two-level quantum system that can be tuned by electromagnetic radiation. For example, spin impurities in solid-state materials have been widely investigated for potential applications. Paramagnetic defects, indeed, such as phosphorus defects in silicon or nitrogen vacancies in diamond, generally have long coherence times, but they are not suitable for the realization of quantum gates because of the difficulties in controlling the qubit–qubit distance during defect implantation [57,58]. In order to overcome this limitation, other spin-based approaches have been taken into account, where both the electronic and nuclear spins of magnetic molecules are exploited.

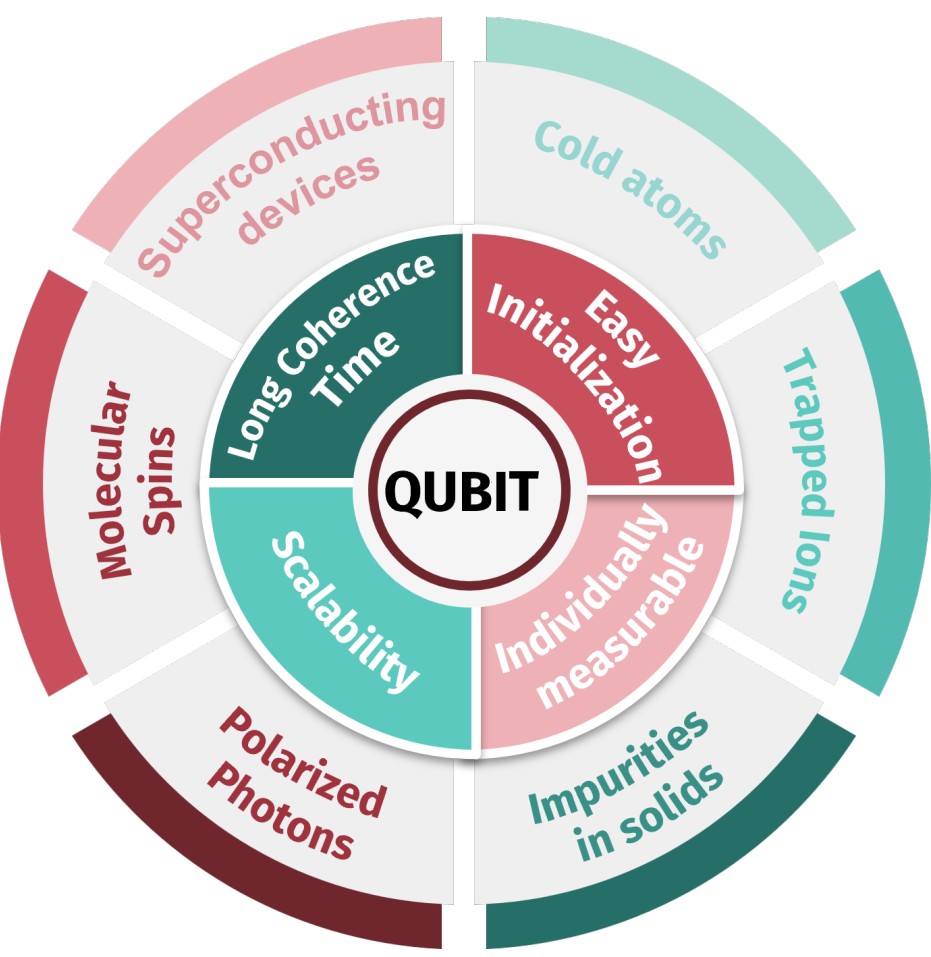

**Figure 7.** Internal circle: the main requirements to have a physical system acting as a qubit. External circle: physical systems used as qubits.

In the past few years, quantum effects in molecular spin systems have been investigated and exploited by scientists of different fields. The first significant development came with the discovery of quantum tunneling in magnetization in the 1990s, which proved that molecule spins are real systems whose quantum properties can be manipulated under controlled circumstance [59]. The nuclear spin of certain atoms or the electronic spin of metal ions in a precise oxidation state, such as a transition metal or a lanthanide in a molecular complex, are some of the emerging qubit candidates. There are advantages and disadvantages in dealing with nuclear or electronic spins. Nuclear spins have long coherence times, are more protected against environmental perturbation (the so-called decoherence), and can be manipulated by Nuclear Magnetic Resonance (NMR) [60]. However, they are not easily tunable, and it is not easy to develop a switchable interaction, which is needed in operating quantum gates. Electronic spins, instead, can be rapidly manipulated with a Electron Paramagnetic Resonance (EPR) [61] that exploits the larger Zeeman effect. The complete balance provides several advantages in using molecule spins. First, they are accessible and controllable using currently known experimental techniques. By fine tuning the spin of Hamiltonian parameters during chemical synthesis, their properties can be correctly tuned. Then, the higher operating temperature avoids the use of refrigerators that are needed for superconductive qubit-based technologies, which give rise to a much broader range of applications. In facts, molecules with low spin values are investigated as fast and switchable dynamical computational units, while molecules with a large spin and magnetic anisotropy showing magnetic bistability at the single molecule level have been proposed for long-term information storage and as sensors [62]. Out of the molecules considered for these applications, coordination compounds receive the most attention due to their adaptability in adjusting parameters like spin manifold, the anisotropy of the g

tensor, and hyperfine coupling. In this part, the requirements for molecule spins to be suitable for qubits will be analyzed in detail, and we will focus on the properties of magnetic compounds, especially coordination compounds. Then, the role of molecular spins in hybrid quantum architectures will be presented, together with a specific application in the case of the Terbium(III) bis-phthalocyaninato (TbPc$_2$) complex.

*4.1. Molecular Chemistry in Quantum Technology*

4.1.1. Coherence Time

The first fundamental requirement for a system to behave as a qubit is to have a long coherence time, that is, the lifetime of the superposition state. The qubit performance is evaluated by extracting two parameters: the spin-lattice relaxation time $T_1$ and the coherence time $T_m$ (which is required to be higher than 100 μs [63]). To extend both, many studies have been carried out. Assuming the temperature to be low enough to exclude spin-lattice relaxation processes, several sources of spin dephasing may be operative. In particular, the following three types of spin–spin interactions can cause decoherence: electronic–electronic dipolar interaction, nuclei hyperfine interaction, and electronic–nuclear dipolar interaction. The electron–electron spin–spin interaction depends on the concentration of the paramagnetic species: when it is low, the dominant interaction is the electronic–nuclear dipolar interaction (which depends on the nuclear magnetic moments and the hyperfine interactions). A promising strategy for enhancing the coherence time involves utilizing atoms with either no or low nuclear magnetic moments. This approach has been successfully employed to exploit paramagnetic impurities in solid-state crystals. In principle, this strategy could also be applied to coordination compounds, but spin active atoms such as N, P, and H are typically found in the ligands of transition metal and lanthanide ions in coordination compounds [64]. Nevertheless, if simpler structures are considered, such as vanadium-based complexes with a nuclear spin-free dithiolene $C_8S_8$, which is introduced in a nuclear spin-free solvent, it is possible to observe a two-order magnitude higher coherence time, as reported in Figure 8. Thus, a longer coherence time can be obtained by using molecule-based qubits, thereby removing the decoherence sources from the environment [65]. Further improvements can be achieved by isolating the spin from the external environment through encapsulation methods. Even though lanthanide-based qubits have a magnetic anisotropy that favors decoherence, longer coherence times were obtained by using fullerenes as cages for the group V element and lanthanide ions. Therefore, more effective findings are expected to be obtained by using transition metal ions. Most of the physical realizations of qubits work at low temperatures, but maintaining long-spin coherence at high temperatures is highly desirable. Indeed, the increase in the operable temperature range of these molecular spin components allows one to avoid the use of dilution refrigerators, which significantly reduces the dimensions of the needed apparatus and the costs of operation. In particular, when above 77 K, liquid nitrogen can be used, thus making the cooling process much easier and cheaper. Decoherence is mostly caused by spin-lattice relaxation times in a high-temperature regime. It was found that this parameter is strongly influenced by the geometry displayed by the specific coordination molecule because of the internal vibrational modes changing.

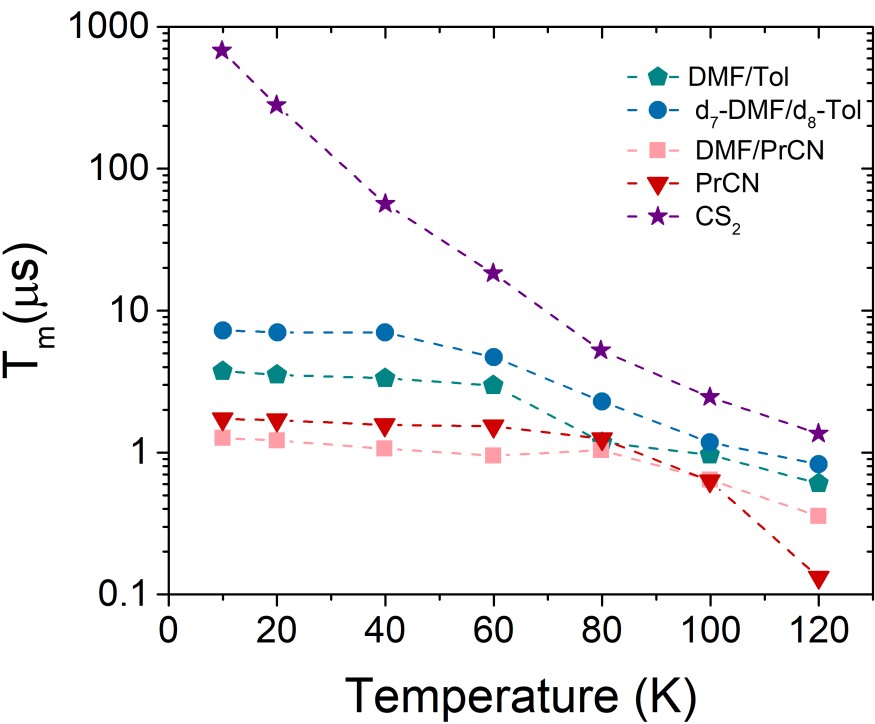

**Figure 8.** A figure adapted from [65]. Revised description: Logarithmic temperature dependence of $T_m$ for $[V(C_8S_8)_3]^{(2-)}$ in protiated, deuterated, and nuclear spin-free solvents, which illustrates the enormous impact of eliminating nuclear spins on the magnitude of the coherence time.

### 4.1.2. Initialization

The initialization is a crucial requirement for allowing the qubit to perform logical operations; all qubits involved in a quantum gate must assume a clearly specified starting state. Many techniques can be used to initialize a qubit, and thermal initialization is among them. The thermal initialization procedure involves applying a sufficiently strong magnetic field during a cooling process. Upon reaching the lowest possible temperature, a spin polarization toward a well-defined spin sub-level of the electronic ground spin state takes place. Since Zeeman energy is sufficient to make the spin levels separated, this strategy proves very effective for electronic spins. On the contrary, this procedure is not applicable to nuclear spins due to the insufficient Zeeman splitting of nuclear energy levels. In such cases, a series of microwaves and radiofrequency pulses are utilized to selectively populate a hyperfine level. The optimal efficiency is achieved by employing frequencies that are typically forbidden for electronic transitions. When employing hybrid nuclear/electronic approaches, initializing hyperfine sub-levels can be achieved through dynamic nuclear polarization (DNP), which exploits laser pulses to induce fluorescent emissions [66]. The pulse intensities depend on the efficiency of the inter-system crossing transitions, thereby enabling the selective population of a specific state within the ground S=1 state. This method was firstly used for initializing spin impurities in solid-state compounds [67]. However, optical initialization has not been as extensively developed for molecular qubits, thus requiring further implementations. Phenomena such as inter-system crossing transitions remain quite unexplored for molecular compounds. Consequently, radiative processes like fluorescence and phosphorescence are needed for the read out.

### 4.1.3. Quantum Gates

The satisfaction of the main requirements for the realization of qubits allows the implementation of quantum gates that can involve one or more structures. Quantum gates are the equivalent of logic gates in classical computing; they are unitary operators that operate on qubits to carry out operations like entangling them, thereby altering their quantum states or carrying out quantum computations. Unitary transformations preserve

the normalization of the quantum state vector, thus ensuring that the total probability of all possible outcomes remains constant. Because of this, any action taken on a quantum state can be reversed by using the inverse operation, which returns the state to its initial state. This reversibility is a crucial feature of quantum computing algorithms as it allows quantum computers to process information efficiently while preserving the integrity of quantum states throughout computations [68]. A typical one-qubit operation is the phase inversion that corresponds to the NOT operation. Two-qubit operations can be performed by using structures in which the state of the second (control) spin influences the dynamics of the first (target) spin. To accomplish this, the two spins must be able to be distinguished either spatially or spectroscopically. Typically, two-qubit gate encoding involves the following steps: system initialization, target spin rotation under various control spin qubit conditions, and system state readout. Having access to a sufficient number of qubits that can serve as both targets and controls enables the execution of sophisticated operations using different gate sequences within an unified platform. Depending on the pulse sequence and rotation chosen, multiple operations like the CNOT and the $\sqrt{\text{iSWAP}}$ can be effectively performed (a schematic representation is depicted in Figure 9). This conditional spin dynamics is made possible thanks to the entanglement of two or more spin centers. Inter-molecule or intra-molecule entanglement can be achieved by manipulating the topology of the spins and their magnetic interactions. The most straightforward method to entangle two spins consists of positioning them within a few nanometers of each other and using the dipolar interaction and other magnetic interactions. The optimal strength of the interaction involved is due to the specific gate implementation. To maintain the single-spin control, the interaction must be larger than the difference in the interaction (either hyperfine or Zeeman) between the individual qubits but not excessively dominant. Additionally, the interaction should not be permanent so as to enable the switchable interaction between qubits to facilitate independent rotations of the two spin qubits. Molecule-based qubits are particularly well suited for meeting this requirement due to the high level of manipulation achievable in molecular chemistry [69,70]. Molecular states can be engineered to utilize auxiliary states, and this is achieved through various strategies like employing differently oriented connected antiferromagnetic rings, utilizing photoactive or redox-active spacers to control qubit interaction, inducing spin crossover transitions or redox isomerism in complexes via external stimuli, and generating spin triple states or radical pairs through specific ligand techniques. Hybrid nuclear/electronic approaches have been utilized to entangle weakly interacting Vanadyl qubits within a discrete molecule structure, and these are mediated by dipolar coupling between their electronic spins [71]. Further research on bi-nuclear molecular systems is currently ongoing, where multi-frequency spectrometers are being employed to enable the separation and/or a switchable interaction between the two qubits. The operating conditions must be set according to the specific case, while the commercial equipment, such as the pulsed electron–electron double resonance (PELDOR), has already been well developed [72]. It is worthwhile to note that magnetic molecules typically have both nuclear and electronic spins; the longer coherence time of nuclear spins can be combined with the easy and fast manipulation and reading out of electronic spins.

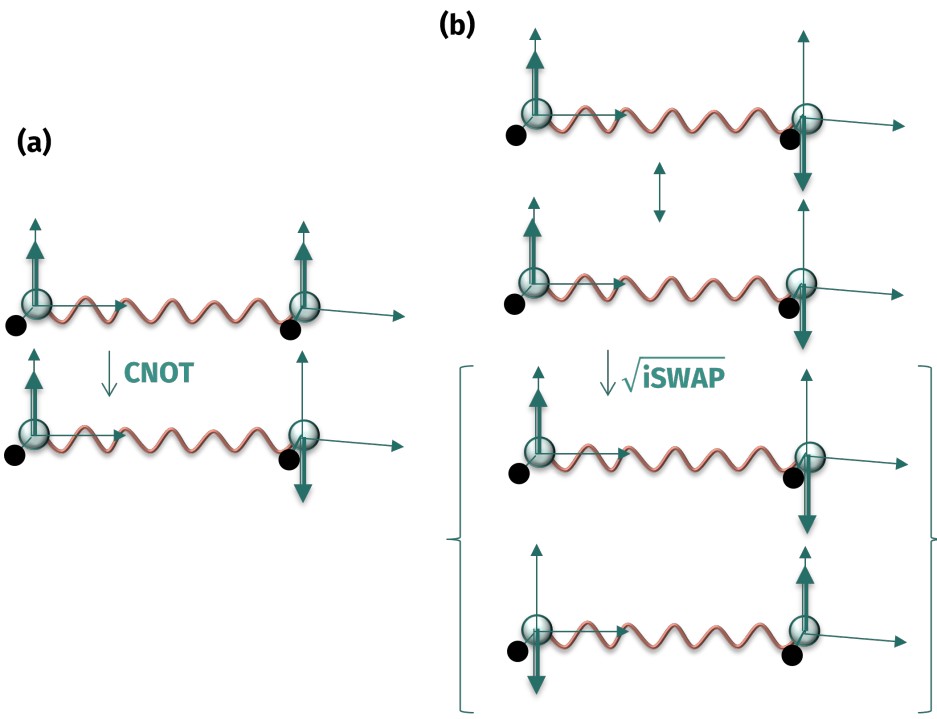

**Figure 9.** Figure adapted from [70]. Adapted description: (**a**) Schematic representation of the effect of the CNOT gate on a pair of qubits, initialized in the computational basis states $|11\rangle$ and $|10\rangle$, respectively. The CNOT flips the target qubit if the control is set to $|1\rangle$. (**b**) Schematic representation of the effect of the $\sqrt{\text{iSWAP}}$ gate on a pair of qubits, initialized in the computational basis state $|10\rangle$. The gate brings $|10\rangle$ to the equal-weight superposition $(|10\rangle + i|01\rangle)/2$.

### 4.1.4. Addressability

The capacity to address single qubits is strictly necessary for the operation and successful performance of quantum gates. In the case of coordination compounds, the objective is to develop efficient techniques for the detection of small magnetic signals, as well as for the precise and selective manipulation and interaction with single or groups of spins while they are protected from the external environment. One of the most used techniques is the placement of a single molecule in a nanocircuit or its deposition on a particular surface, which is followed by detection using a probe with sufficient lateral resolution. The spin of a molecule can be strongly influenced by the external ambient. The robustness of the molecule is an important feature for these applications. The meticulous study of isolated molecules on surfaces has been widely conducted using different methods, including Scanning Tunneling Microscopy (STM) and X-ray spectroscopy. As will be discussed below, molecules that retain their magnetic properties after the deposition on specific surfaces can be also used as sensors. Different approaches for detecting tiny magnetic signals have been explored. One of them involves using nano-SQUIDs made with carbon nanotubes [73]. However, the magnetic coupling between the molecules and the sensor represents a significant limitation. The use of scanning probes, like tunneling tips, appears to hold greater promise due to their ability to localize and read single atoms and molecules. Nevertheless, there is still a great deal to understand about data interpretation, especially since magnetic features seem to be delocalized in the presence of organic ligands. It has been reported that tunnel junctions may host a single molecule; however, understanding how the charge current from the leads perturbs the magnetic state of the molecules remains still unclear. Another potential read-out scheme involves utilizing quantum dots, whose conductivity is impacted by the spin state of the magnetic center coupled with the device. Quantum dots can be fabricated using carbon nanotubes, graphene, or organic radicals [74–76]. Various strategies can be considered for implementing a two-qubit gate with molecular spintronic devices. For example, a carbon nanotube (CNT) could accommodate two or more molecules, and there

are multiple gate electrodes that have the ability to turn on and off molecular interactions. Then, spin manipulation can be conducted by addressing each spin through microwave pulse sequences. Reliability is a critical requirement for these operational quantum devices, thus making necessary the establishment of effective quantum error correction techniques. In the field of computer science, addressing the challenge of error correction is well recognized. Accidental flips of a qubit during a quantum operation can occur, thereby making the entire procedure incorrect. To reduce and fix the errors, the qubit can be encoded across multiple processors, typically three, and the majority rule can be applied [77]. According to this rule, if the chance of an unintended flip is sufficiently low, two of three qubits will remain in the correct state, while one will contain an error. This error-correction protocol opens another technological challenge. In order to scale quantum computing systems, it is fundamental to have arrays of similar devices for the manipulation and read outs that can function simultaneously. However, the success rate in the fabrication of tunnel junctions or CMOS-compatible quantum devices below 10 nm remains notably low.

### 4.1.5. Scalability

Parallel to the ongoing size reduction in electronics, scalability is another important step to consider in the realization of more complex quantum architectures. Indeed, the implementation of two-qubit gates represents only a basic attempt to exploit the total potential of quantum algorithms' complexity. Physical systems can be made scalable using two different approaches: vertical and horizontal scalability. Vertical scalability involves incorporating additional electronic transitions associated with a ground state $S > 1/2$ or when using hyperfine coupling. However, this approach presents a low number of available states. On the other hand, horizontal scalability consists of increasing the number of interconnected qubits in bi-dimensional or tri-dimensional arrays. Several prototypes following this approach have been developed [78,79]. The aim is to intentionally entangle finite ensembles of interacting qubits in a precisely engineered molecular architecture, thus yielding to an efficient scale up of the system. One major challenge is the complexity of scaling these systems when dealing with ensembles. Stated differently, addressing and resolving issues, including relative error correction, requires a growing number of spins as the data increase. Consequently, this architecture may not be suitable for all applications due to rapidly escalating spin requirements. A feasible initial approach could involve developing small-scale quantum computers tailored to handle specific tasks without an excessive need for classical computing power. In practice, this entails utilizing molecular derivatives composed of well-defined, compact spin clusters. These clusters can resolve specific problems dynamically. However, it is important to note that this area is largely unexplored and demands collaborative efforts between chemists and physicists.

### 4.2. Molecular Spins in Hybrid Quantum Architectures

The ability to manipulate and read out an arbitrary spin state is also an important prerequisite for using molecular systems in quantum information processes, like exchanging quantum information between resistors and photons. Molecules offer a broad spectrum of frequencies for a good coupling with photons: nuclear spins are active at radio frequency, while electronic spins cover the microwave range. Additionally, there are molecules that are active in the visible range. Coherent spin-photon states can be achieved in the strong coupling regime, in which the coupling is stronger than the decoherence mechanisms of both spins and photon systems. Conventional superconducting resonators can be used to achieve strong coupling. High-$T_c$ superconducting planar resonators show excellent performance even at finite temperatures and in the presence of strong magnetic fields. An alternative approach involves locally enhancing microwave radiation intensity through nanostructured superconducting strips, thus enabling a strong coupling regime, even with a single spin. The utilization of microwave photons allows for the integration of molecular spins into hybrid quantum devices [80]. Typically, superconducting circuits are employed to facilitate the movement of quantum data between quantum registers and memories [81].

Magnetic Material–Superconductor Coupling

The coupling between magnetic materials and superconductors significantly influences the optimal performance of quantum devices. Molecular qubits can be integrated into quantum circuits by the use of hybrid molecular–superconductor designs, as was previously mentioned. At the nanoscale, local bound states are created within the superconductive band gap as a result of single spins interacting with superconducting substrates; this phenomenon seems promising for the creation of Majorana modes. Scientific research and several companies are investing and investigating these systems in search of Majorana modes. These modes are very attractive qubit candidates because of their topological robustness. Furthermore, positioning molecules on a substrate enables the individual addressing of these molecules. Magnetic molecules situated on a superconductive substrate exhibit interesting properties. The deposition of single-molecule magnets (SMMs) with magnetic memory in a single layer on Pb(111) has advanced the exploration of the hybrid materials resulting from the combination of molecules and superconductors. SMMs are finite molecular species characterized by a gradual relaxation of their magnetization. They maintain magnetization even at low temperature, similar to bulk materials. This phenomenon arises from the bistability of the ground state, where the reversal of the magnetization is hindered by an energy barrier. SMMs behave as nanosized magnets, demonstrating hysteresis and the quantum tunneling of magnetization because of the interplay between a large molecular spin and an easily oriented axis of anisotropy. SMMs have been extensively investigated not only for their potential application as qubits in quantum computing and in quantum information processing, but also as sensors at the nanoscale. It has been shown that the magnetization dynamics of SMM complexes are affected by Pb transition to the superconducting state, thereby leading to a quantum tunneling of the magnetization that causes the magnetic state to change from a blocked to a resonant domain [82]. An indirect procedure that detects the superconducting state involves depositing an inorganic ferromagnetic layer and monitoring its magnetic evolution using methods such as X-ray adsorption or magneto-optical techniques. However, this approach is limited by the intrinsic spatial resolution of the probe and the mediated response due to the correlation of the ferromagnetic layer [83]. On the contrary, establishing contact between the nanometer-sized magnetic molecules and the superconductive surface provides an independent response with nanometric resolution.

### 4.3. TbPc$_2$ as the Local Sensor of the Superconductive Phase

Molecules that present permanent magnetic properties can be used as sensors. In particular, SMMs with a magnetic memory that are deposited on Pb(111) substrates find several applications in the quantum sensing field [82]. An innovative coordination compound that presents these features is TbPc$_2$. It is a double-decker system composed of two phthalocyanine (Pc) molecules that coordinate a terbium (Tb) ion. The chemical and structural stability of the compound allows for its processing in an ultra-high vacuum and from a solution [84]. Additionally, due to its exceptional magnetic anisotropy, the primary relaxation mechanism of the electronic magnetic moment at low temperatures is through the tunneling of the anisotropic energy barrier [85,86]. These features make it highly suitable for single-spin detection and manipulation [87]. Importantly, the nuclear dynamics can be investigated by measuring the electronic one, which can be directly detected at the single molecule levels through transport measurements owing to their strong interconnection. Integrating a single TbPc$_2$ molecule into a nanometric junction leads to the formation of a three-terminal single-molecule magnet transistor. Studies have confirmed that the hopping electrons primarily engage with Pc ligands, while the unpaired electrons on Pc interact with the Tb magnetic moment. This validation confirms the possibility of controlling nuclear spin dynamics by investigating electron transport mechanisms [88]. Furthermore, the manipulation of nuclear spin can be achieved by applying an electric field, thus utilizing the Stark effect on the hyperfine interaction. These unique properties make TbPc$_2$ the suitable candidate for implementing the Grover algorithm that was proposed to find an element in an unsorted list [89]. As previously mentioned, TbPc$_2$ exhibits a strong

uniaxial anisotropy, with the easy axis of magnetization perpendicular to the Pc planes [90]. At liquid helium temperatures and in bulk form, the substantial energy barrier between the ground doublet states inhibits the reversal of magnetization, thereby rendering TbPc$_2$ a single-molecule magnet. This compound is particularly sensitive to the interaction with the substrate. Remarkable effects have been observed in the magnetic hysteresis loop of the TbPc$_2$ films [91]. The hysteresis is suppressed when the molecules interact with metals or is amplified when using decoupling layers. This sensitivity was exploited to investigate the magnetization behavior of TbPc$_2$ at the interface with Pb(111) and throughout its superconducting transition while varying the temperature and the applied magnetic field. For this investigation, a sub-monolayer film of TbPc$_2$ molecules was deposited through a thermal sublimation on the Pb(111) surface [92].

Analysis of the normalized X-ray Magnetic Circular Dichroism (XMCD) spectrum as a function of magnetic fields revealed different behaviors [93]. Above the critical temperature and field of Pb, the expected opening attributed to the SMM behavior was quenched across all temperatures, thereby making TbPc$_2$ behave as a paramagnet in this regime [94,95]. However, when Pb goes in the superconductive state, hysteresis appears in the XMDC spectrum. This effect occurs since the normal and superconducting domains have distinct topologies when the magnetic flux enters or leaves the superconductor. Type I superconductors are known to show an intermediate state below their critical field and temperature, where superconducting and normal domain phases coexist at the micrometric scale. When decreasing the field below $H_c$, i.e., during the magnetic field expulsion phase, the magnetization curve of the Pb crystal shows small absolute magnetization values by having little impact on the magnetization of TbPc$_2$. The trend of TbPc$_2$ magnetization deviates only slightly from linearity below $H_c$ under these conditions. In this scenario, the superconducting (s) and normal (n) domains of the Pb crystal exhibit a laminar topology. During the magnetic field expulsion, the sample can expel the magnetic flux only through the n domains, which decrease in size as the magnetic field intensity is lowered until reaching zero field. At zero field, the substrate is completely in the superconducting Meissner state, and the magnetic flux is entirely excluded [96]. It is important to note that, in the laminar topology of the intermediate state, the magnetic flux is never fully screened by the superconductor until the magnetic field intensity reaches zero. When increasing the external magnetic field intensity from zero, the bulk of the sample does not allow magnetic flux penetration, and it remains in an almost diamagnetic state until a certain field $H_p$. Magnetic flux cannot move toward the center of the Pb disks during this phase; it can only enter at their edges [97]. This behavior explains the overall zero XMDC values detected on the TbPc$_2$ layer when the field increases from zero to $H_p$. Above $H_p$, the magnetic flux penetrates through the n domains, which expand from the edge over the whole sample, thereby forming a tubular topology and favoring a hexagonal symmetry. During this phase, TbPc$_2$ molecules, acting as sensors, make the XMCD signal increase thanks to the expansion of the n domains. It is essential to remark that, in the intermediate state, the magnetic field intensity in normal regions always equals $H_c$, and the observed increase in XMCD intensity is a result of the average over n domains with increasing extension (see Figure 10).

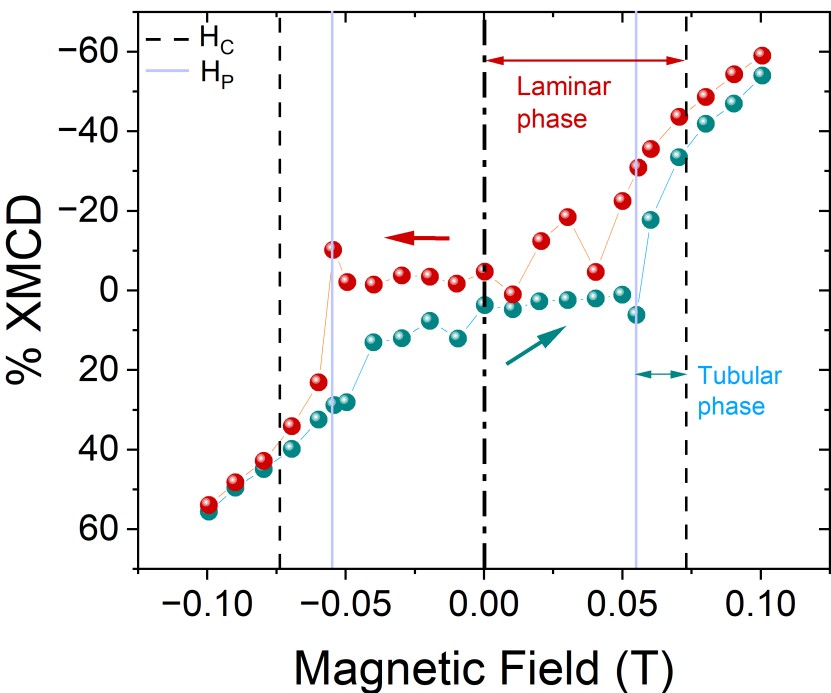

**Figure 10.** Figure adapted from [96]. Adapted description: Magnetization curves at 2 K and $\theta = 0°$ of monolayer TbPc$_2$ on Pb(111) within the superconductor's critical field ($H_C$). For increasing magnetic field intensity within a given field $H_p$, the complete magnetic field screening effect of Pb is evident. The direction of the field scan is shown by the arrows.

*4.4. Summary*

The Second Quantum Revolution has had a substantial impact on numerous fields, notably on quantum sensing. The efficacy of quantum devices has been profoundly affected by the interplay between magnetic materials and superconductors. The integration of molecular qubits into quantum circuits has been made possible through hybrid molecular–superconductor architectures. Molecules with magnetic properties unaffected by the substrate on which they are deposited can serve as sensors for superconductive phases. The utilization of a single layer of magnetic molecules not only enhances sensitivity to the transition into the superconducting state, but also enables detection of the topological features within the superconducting domains. TbPc$_2$, given its strong out-of-plane magnetic anisotropy and the capacity to form highly oriented films, stands as a remarkable example. These molecules exhibit exceptional sensitivity to variations in local magnetic flux, thereby enabling them to be valuable tools for detecting and investigating superconductivity. This method represents a significant advancement compared to conventional approaches based on SQUID and magneto-optic methods. Unlike bulk measurements, this method operates at a local level, thereby enabling the examination of ultra-thin films, which is particularly relevant for studies related to topological superconductivity. Moreover, each molecule operates as an independent probe at the nanometric scale, thus overcoming the resolution limitations of both optical probes and magnetic correlations found in the ferromagnetic indicator disks of magneto-optic techniques. This characteristic is highly pertinent for investigations into nanoscale superconductivity. Furthermore, the intrinsic topology of the intermediate state in superconductors has been demonstrated to induce a hysteresis effect in an ensemble of paramagnetic molecules. This last outcome holds a great potential for various technological applications. These findings are particularly relevant to fields utilizing hybrid molecular/superconductor systems in macroscopic devices like resonators, as well as for detecting localized states occurring at the interface between individual spins and superconducting surfaces. This highlights the significant role of molecules in advancing quantum technologies.

## 5. Harnessing Quantum Complexity and Implications for Potential Industrial Applications

In this review, we delve into the transformative potential of the Second Quantum Revolution, which promises groundbreaking advancements in technology, computation, and communication. We discover the intrinsic complexity of quantum systems and investigate the significant implications for commercial applications as we navigate through the complexities of quantum mechanics. This section highlights the approaches and techniques for utilizing the quantum complexity included in these systems and clarifies how they might affect different industries. We analyze how the laws of quantum physics are changing the face of technology and opening the way to previously unheard-of breakthroughs in fields ranging from materials science and sensing to quantum computing and cryptography. We aim to clarify the advantages and disadvantages of incorporating quantum technologies into industrial procedures and emphasize their revolutionary potential for reshaping the industry through an examination of current advancements and theoretical frameworks.

The interest toward quantum technologies and quantum computers has increased in the last years, not only in the academic framework, but also in the industrial one. This seems clear from the analysis of investment in these sectors. As reported by McKinsey & Company in their Quantum Technology Monitor, which was published in April 2023, the amount of investment in quantum technology start-ups has increased by 75% in the last year (see Figure 11 [98]). E. Gibney also reported the total amount of deals per year in the different sectors of quantum technologies in her Quantum Gold Rush [99].

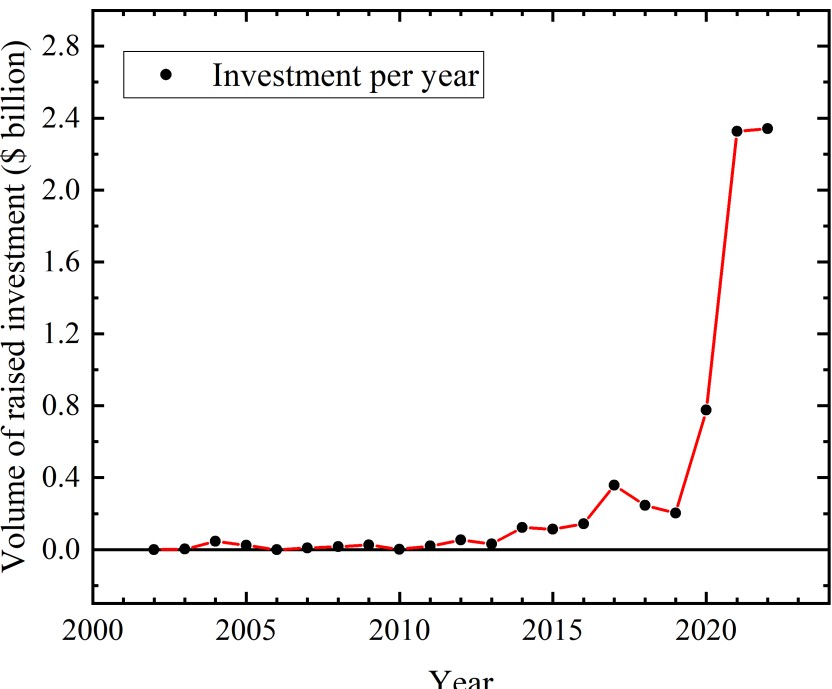

**Figure 11.** Figure extracted from the Quantum Technology Monitor 2023, McKinsey & Company [98]. Original description: Based on public investment data recorded in PitchBook. Actually investment is actually higher. Source: PitchBook.

The amount of investment was about USD 2.35 billion. The main reason for all this investment was the promise of quantum technologies in solving problems that modern computation is unable to solve [100,101]. These problems are connected to the calculations of exponential complexity that can be simplified to polynomial complexity using quantum computers. This potential opens the door to the fundamental industrial applications of quantum computation and quantum technologies in general. These are connected to the fields of cybersecurity, theoretical simulation, and the resolution of complex mathematical problems [102].

In the case of cybersecurity, quantum computing is the science that sets the end of classical cryptography, as well as the evolution of a new way through which to secure communication protocols [103]. The research in this field has been mainly supported by governments and organizations connected to the storage of private data, like banks or societies that manage clouds, due to the risks connected to quantum decryption [101].

In the pharmaceutical and chemical sectors, quantum computers can revolutionize the simulation of complex molecular processes. The ability to model atomic interactions with unprecedented precision accelerates the development of new materials and drugs, thus reducing the time and costs associated with research and development [104].

In the financial sector, quantum computers can optimize predictive analysis algorithms and solve complex optimization problems, improving portfolio management and optimizing trading strategies. This advanced computing power can open new opportunities for financial data analysis and enhance market understanding [105,106].

Logistics and supply chain management are other areas where quantum computers can bring substantial improvements. The complexity of distribution networks and stock management can be more efficiently optimized, thereby reducing costs and improving customer satisfaction through more accurate planning [107].

Quantum computers can be also used to solve optimization problems in the energy sector, enhancing the energy efficiency of industrial systems. Indeed, quantum computers have the potential to address complex optimization challenges in the energy sector, thereby leading to improved efficiency, cost savings, and more sustainable energy solutions.

Here are some of the key aspects and potential applications of quantum energy optimization:

- Grid Optimization: Quantum algorithms can be employed to optimize the operation and control of power grids, thus ensuring efficient energy distribution and minimizing transmission losses [108].
- Renewable Energy Integration: Quantum computing can assist in optimizing the integration of renewable energy sources, such as solar and wind, into existing energy grids. This involves addressing the variability and intermittency of renewable sources [109].
- Resource Allocation: Quantum algorithms can optimize the allocation of energy resources, including the scheduling of power generation from various sources to meet demand while considering factors like cost and environmental impact [110].
- Energy Storage Management: Quantum computing may enhance the optimization of energy storage systems by helping the determination of the optimal locations and capacities for energy storage facilities, as well as by delivering the most efficient use of stored energy [111].
- Smart Grids and Demand Response: Quantum algorithms can contribute to the development of smarter energy grids, enabling better demand–response mechanisms and the efficient utilization of energy resources based on real-time demand fluctuations [112].
- Carbon Emission Reduction: Quantum optimization can be applied to minimize carbon emissions by optimizing the energy mix, thus reducing reliance on fossil fuels and identifying cleaner energy alternatives [109].
- Supply Chain Optimization: Quantum computing can optimize the supply chain for energy-related components, thereby leading to improved efficiency in manufacturing, transportation, and maintenance processes [113].
- Exploration of New Materials: Quantum computers can aid in the exploration of new materials for energy storage and conversion, thus potentially accelerating the development of advanced energy technologies [114].

It is important to note that while the potential applications are promising, the practical implementation of quantum algorithms for energy optimization is still an area of active research. The development of scalable and error-tolerant quantum computers is crucial for realizing the full potential of quantum optimization in the energy sector. As the field

progresses, more insights and breakthroughs are expected to shape the future of quantum energy optimization. In the field of artificial intelligence, quantum computers can expedite the machine learning process, enabling the training of more complex models and handling colossal datasets. This paves the way for significant developments in applications such as speech recognition, computer vision, and data analysis. In conclusion, quantum computers open up scenarios of innovation and progress in various industrial sectors, which is promising for redefining the dynamics of information technologies and radically changing how we tackle complex computational problems. The growing interest and substantial investments reflect the enthusiasm and awareness of their potential impact on global industries. The simple overview proposed for the application of quantum computers can help to understand how industries are interested in investing in this technology. In summary, the advantages reside in the possibility of solving difficult computational problems in a reduced amount of time. As a consequence, one can think of increasing the complexity of these problems and taking more benefits from their solutions (see Figure 12). Otherwise, at the moment, the use of quantum computers is a future prospect due to the problems connected to the scalability of quantum circuits and the errors connected to the measurement of the coding results. In fact, in the last year, investments have moved from building new start-ups to powering up the existing ones (see Figure 13 [98,115]). This is a consequence of a society that has found the frame of applicability of this new technology but needs a technological evolution to gain benefits from it. For this reason, the majority of investment is moving from the investigation of new regions of applicability of quantum computers to the necessity of obtaining more stable quantum bits to ameliorate the problems of our interest [98,115].

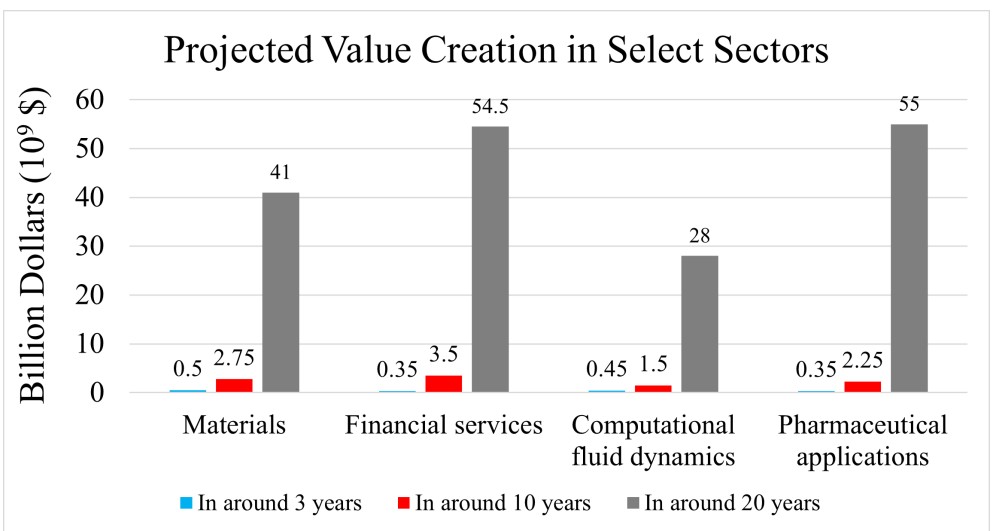

**Figure 12.** Projected value creation in the different sectors extracted from Where Will Quantum Computers Create Value—and When? [115] Original description: Source: BCG Analysis.

Since these aspects are largely discussed and are quite well known, we discuss here a new frame where this technology can be successfully applied, i.e., in law enforcement.

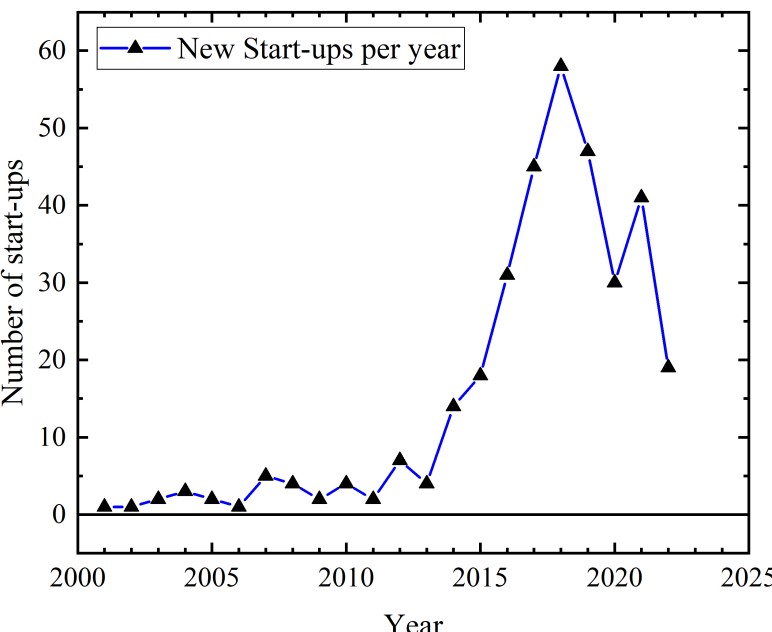

**Figure 13.** Number of new start-ups per year, as extracted from the Quantum Technology Monitor 2023, McKinsey & Company [98] Original description: Source: Crunchbase; PitchBook.

### *5.1. Quantum Technologies and Law Enforcement*

It is worth noting that quantum sensors and communications can also be exploited for law enforcement. However, it is important to take into account the risk that criminals could also have access to them. We will now outline the risks and provide specific actions that law enforcement should take to mitigate them [116,117]. Overall, we would like to emphasize the importance of law enforcement preparation in the face of emerging developments in quantum technologies by highlighting both the potential challenges and opportunities that these technologies may bring to the realms of security and law enforcement (see Figure 14) [118,119]. Let us start by looking at the growing impact of artificial intelligence (AI), machine learning, and data analysis technologies, particularly in the context of law enforcement and criminal investigations because of the ease with which large volumes of data can now be processed and analyzed. In particular, a new area of multidisciplinary research, called quantum machine learning, has drawn a great deal of interest lately [120–122]. It provides innovative predictive tools and enables the analysis of complex and huge datasets [123]. Furthermore, biometrics and computer vision for sensors or image analytics, together with pattern recognition techniques that identify offenders or localize crimes, are important applications [124–126]. As mentioned previously, cybercriminals can easily take advantage of these techniques through the exploitation of AI-specific vulnerabilities in software systems and AI-based malware, which they can use to increase their capacity for password guessing and hacking [127]. Proven quantum algorithms have shown exponential speeding up for crucial machine learning and simple data processing tasks, especially when a fault-tolerant, universal quantum computer is developed. Quantum Basic Linear Algebra Subroutines (QBLAS), a set of fundamental routines, could significantly impact machine learning and statistical data analysis. Some examples are Grover's algorithm for database search tasks (quadratic speed up); the HHL algorithm for tasks like matrix inversion and finding eigenvalues and vectors (quadratic speed ups); and the quantum Fourier transform for data processing (exponential speed up). Even current NISQ (Noisy Intermediate-Scale Quantum) devices are useful for approximation algorithms in these fields [128]. The combination of hybrid quantum–classical methodologies and approximate algorithms executed on NISQ devices has demonstrated promise for achieving a quantum advantage in machine learning. Experiments with quantum neural networks, quantum kernel-based machine learning, and variational optimization algorithms like the

quantum approximate optimization algorithm (QAOA) or quantum annealing demonstrate potential in machine learning, optimization tasks, and data analysis [128]. Moreover, quantum technology's advantages extend beyond improved machine learning performances, which potentially involve a broader range of applications. There is a discussion to be had about quantum accelerators that could parallel the role of GPUs in classical machine learning. Additionally, quantum algorithms and data representations might lead to better generalization behavior by addressing challenges like over-fitting in deep learning. While these developments create opportunities, the emerging quantum advantage, often lacking theoretical guarantees, should be carefully monitored in the near and mid-term.

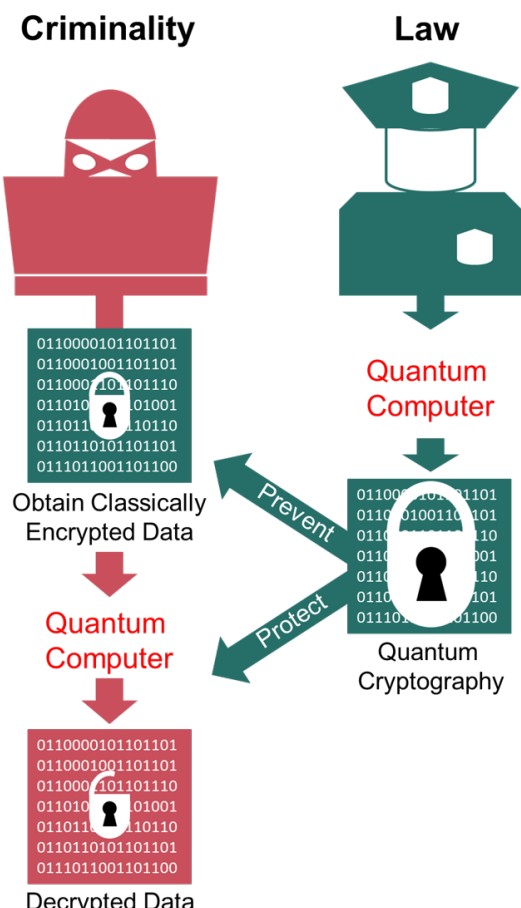

**Figure 14.** A simple example of competition between criminality and law in the use of quantum technology. At the moment, the main strategy for criminality is to obtain classically encrypted data that can be decrypted after the introduction of quantum computers. To avoid this scenario, law enforcement is also working on implementing quantum cryptography to protect data in the quantum era.

Quantum key distribution involves exchanging a secret key between a sender and a receiver using quantum technology. Information is transmitted through a quantum channel and random bits are encoded using entangled quantum particles such as photons. The main benefit is the use of quantum mechanics, which ensures intrinsic security against manipulation or eavesdropping as any interaction with the quantum particles changes their state. This strategy guarantees extremely secure key exchange protocols, thereby protecting critical infrastructures, sensitive data, and communications [129,130]. Several sectors, including law enforcement, are expected to undergo substantial changes due to the application of quantum sensors and quantum metrology in the rapidly developing field of new quantum technologies. Quantum sensors, which can be considered the focus of this

report, have already gained high levels of measurement efficiency in experimental activities [131]. These sensors enhance the accuracy, precision, and sensitivity of measurements, thus giving rise to innovative applications that can impact law enforcement procedures and operations, such as surveillance, detection methods, and situational decision making based on sensor data [132]. Quantum sensing has undergone more extensive development compared to quantum computers, despite still being in its initial stages. Recognized for its transformative potential in the defense security sector, it is recommended that law enforcement agencies and internal security institutions pay attention to these advancements. Some capabilities are already attainable, with the likelihood of additional technologies becoming available in the near future [124]. The first tangible impact of quantum technologies that underlie qubit development may manifest through enhanced measurement technologies before the realization of a fully functional quantum computer. In the context of forensic crime scene investigations, which heavily rely on various forensic sensors, modern quantum technologies could introduce new types of sensors with the potential to replace costly forensic laboratory tests. These quantum sensors could be faster, smaller, more cost-effective, and potentially even more effective when deployed at a crime scene [125]. Examples include nanoparticle-based biosensors for biological traces and proposed sensors for the environmental screening of chemicals and toxicological substances. The goal is to increase the sensitivity and specificity of forensic tests, thus offering potential advancements in crime scene analysis [126].

*5.2. Summary*

However, despite the revolutionary promises, quantum computer technology is still in its infancy and faces significant challenges. Quantum decoherence and susceptibility to errors require the development of advanced error correction techniques to ensure the reliability of computations. Numerous companies and research institutions are working tirelessly to overcome these challenges and make quantum computers more stable and accessible [123,133]. In conclusion, quantum computers represent an exciting prospect for technological and industrial evolution. Their potential applications are so diverse and transformative that their widespread commercial adoption could usher in a new era of innovation and progress. Despite current challenges, enthusiasm and investments continue to grow, thereby fueling the hope for a future in which quantum computers become a key resource for addressing the complex challenges of our society and industry [134,135].

## 6. Conclusions and Perspectives

While technical and theoretical challenges persist, the Second Quantum Revolution marks a fundamental turning point that is already shaping the future of technology and science, with potential revolutionary applications to arrive in the coming decades. Indeed, the Second Quantum Revolution marks a transformative era in technology through leveraging the principles of QM for practical applications. Quantum computing, communication, and sensing advancements hold the potential to revolutionize industries, thereby enabling unprecedented computational power, secure information transmissions, and highly precise measurements. The perspectives and future improvements of these mentioned applications are interconnected and often reinforce each other, and they contribute to the overall advancement of quantum science and technology. Understanding these interrelations provides a holistic perspective on how advancements in one aspect of the Second Quantum Revolution can impact and complement progress in other areas, thereby fostering a more integrated and synergistic development of quantum science and technology.

We also note that the Second Quantum Revolution has the potential to significantly impact teaching and formation (educational training) across various levels, from undergraduate studies to advanced research. Indeed, it is likely to reshape educational approaches by integrating quantum concepts into curricula, providing hands-on experiences and fostering interdisciplinary collaboration. This evolution can empower students to contribute meaningfully to the ongoing developments in quantum science and technology.

Finally, we would like to stress that the Second Quantum Revolution introduces a range of ethical considerations that demand careful attention. Entanglement introduces ethical considerations primarily in the context of quantum communication. The potential for secure data transmission using entanglement raises questions about privacy, surveillance, and the responsible use of this technology. Ethical discussions revolve around ensuring data ownership, preventing unauthorized access, and establishing guidelines for the ethical deployment of quantum communication systems. Concerns include ensuring the responsible and secure use of quantum computing, particularly in regard to the potential breaches of privacy through advancements in decryption capabilities. Ethical frameworks must be established to address the social and economic impacts, thus preventing the exacerbation of inequalities and job displacement. Additionally, there is a need to regulate the development and deployment of quantum technologies in military applications to prevent unintended consequences and to ensure global stability. Environmental sustainability and minimizing the ecological footprint of quantum technologies are also essential ethical considerations. Striking a balance between protecting intellectual property rights and fostering open access to quantum knowledge is crucial for equitable progress. Overall, ethical guidelines must be proactively developed to navigate the societal implications of the Second Quantum Revolution. Therefore, the ethical discourse is integral to guiding the development and deployment of quantum technologies in a manner that is consistent with societal values.

**Author Contributions:** Conceptualization, K.I., L.V. and C.N.; methodology, K.I., L.V., M.M. and C.N.; investigation, K.I., L.V., V.L., A.M., B.M., M.M. and C.N.; resources, C.N.; writing—original draft preparation, K.I., L.V., V.L., A.M., B.M., M.M. and C.N.; writing—review and editing, K.I., L.V., V.L., A.M., B.M., M.M. and C.N.; visualization, V.L., A.M. and B.M.; supervision, C.N. All authors have read and agreed to the published version of the manuscript.

**Funding:** This research received no external funding.

**Conflicts of Interest:** The authors declare no conflicts of interest.

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
