# Peer review of "The Second Quantum Revolution: Unexplored Facts and Latest News"

_encyclopedia, doi:10.3390/encyclopedia4020040_

Round 1

Reviewer 1 Report

Comments and Suggestions for Authors

The manuscript is a detailed review about quantum mechanics, quantum entanglement. Also the authors gave detailed information about how to construct qubit technologies in section 4.

The manuscript will be suitable for publication when the following comments are considered.

In the introduction part the authors may give some information about “Everett’s many-world interpretation of quantum mechanics”. In some sense the existence of other worlds can explain the quantum entanglement.

In the page 1 line 28 there is a question mark in reference parenthesis.

In the page 2 line 68 the author claims the crisis in classical physics starts at 21st century. But it is mistaken, the crisis began at 20th century.

In the page 6, Bell theorem was written, the authors may explain, Bell inequality and mention 2022 Nobel prize which is about the experimental evidence of violation of Bell inequality.

In the page 7 line 243, there is a colon.

There is a typo in ref 48.

In section 3.2.5 the authors mentioned some applications of entanglement, recently there are some works about the usage of entangled states as fuel for the quantum heat engines.

A reference on it:

Ozaydin, F.; Sarkar, R.; Bayrakci, V.; Bayındır, C.; Altintas, A.A.; Müstecaplıoglu, Ö.E. Engineering Four-Qubit Fuel States for Protecting Quantum Thermalization Machine from Decoherence. Information 2024, 15, 35

Comments on the Quality of English Language

The English quality is good and the manuscript is understandable.  

Reviewer 2 Report

Comments and Suggestions for Authors

i)

What is Copenhagen interpretation. What are the other interpretations of measuring paradox?

ii)

It would help to define a qubit (mathematical) as well as an entangled quilt.

iii)

What is a quantum gate and what is its relation to unitary operator, means quantum computation is reversible.

vi)

Page 2

principles of QM to perform calculations at 43 speeds and processing power unimaginable for conventional computers.  What does it mean? Since there are serious constraints, the speed up seems only quadratic for most problems.

v)

Section 5. Harnessing quantum complexity and implications for potential industrial applications 

seems little bit out of context, or the transition from your narrative is quite abrupt.

Minor, page one citation for point three Schroedinger's Wave Equation is missing

Round 2

Reviewer 1 Report

Comments and Suggestions for Authors

In the manuscript the authors completed almost all of my comments. Then it will be suitable for publication. 

Author Response

We thank the referee for the comments. We really appreciated his/her contributions. 

Reviewer 2 Report

Comments and Suggestions for Authors

Comments 4: Page 2

The naive assumption that quantum algorithms give an exponential speed up is not correct, and you should mention it. (Otherwise the the uninformed reader can get a wrong idea)

Even Shors algorithm gives an exponential speed up, it is not assumed that factorization is NP-Completete. We can determine the period of a periodic function exponentially faster using QFT. For NP-Completet algorithms we "only" get quadratic speed up (Grover's algorithm).  And it is proven (Bennet et all 1997) mathematically that we cannot get better.

Bennett, C. H., Bernstein, E., Brassard, G. and Vazirani, U. (1997). Strengths and weaknesses of quantum computing, URL http://www.citebase.org/ abstract?id=oai:arXiv.org:quant-ph/9701001.

Round 3

Reviewer 2 Report

Comments and Suggestions for Authors

Paper is ready to be published